# Prompts to Proxies:
# Emulating Human Preferences via a Compact LLM Ensemble

## Abstract

Large language models are increasingly used as proxies for human subjects in social science research, yet external validity requires that synthetic agents faithfully reflect the preferences of target human populations. We introduce *preference reconstruction theory*, a framework that formalizes preference alignment as a representation learning problem: constructing a functional basis of proxy agents and recovering population preferences through weighted aggregation. We implement this via *Prompts to Proxies* (`P2P`), a modular two-stage system. Stage 1 uses structured prompting with entropy-based adaptive sampling to construct a diverse agent pool spanning the latent preference space. Stage 2 employs L1-regularized regression to select a compact agent ensemble whose aggregate response distributions align with observed data from the target population. `P2P` requires no finetuning and no access to sensitive demographic data, incurring only API inference costs. We validate the approach on 14 waves of the American Trends Panel, achieving an average test MSE of 0.014 across diverse topics at approximately 0.8 USD per survey. We additionally test it on the World Values Survey, demonstrating its potential to generalize across locales. When stress-tested against an SFT-aligned baseline, `P2P` achieves competitive performance using less than 3% of the training data.

## 1. Introduction

Large language models (LLMs) hold considerable promise as proxies for human subjects in social science research. From agent-based modeling of community dynamics (Park et al., 2023; Zhao et al., 2024a), to simulating survey responses (Argyle et al., 2023; Park et al., 2024) and prototyping policy interventions (Hou et al., 2025), synthetic agents afford scalability, reproducibility, and cost advantages unmatched by traditional human samples (Horton, 2023; Manning et al., 2024). However, realizing this promise hinges on solving a fundamental validity challenge: the preferences expressed by synthetic agents must faithfully reflect those of the target human population, at least when restricted to the topic domain. Without such alignment, conclusions drawn from synthetic experiments lack external validity (Shadish et al., 2002), limiting the scientific value of LLM-based social simulation.

This alignment challenge departs from conventional AI safety framing centered on steering a model toward a single set of desirable values, but shares motivation with the emerging notion of pluralistic alignment (Chen et al., 2025; Feng et al., 2024; Sorensen et al., 2024b)[1]. The goal here is *distributional fidelity*—ensuring that an ensemble of synthetic agents reproduces the heterogeneity of preference patterns observed in a target population. In reality, this challenge is compounded by an invisible resource barrier: while well-resourced industrial labs can deploy supervised fine-tuning to train personalized models on large annotated corpora, small academic institutions and independent researchers often face limited compute, limited data, and ethical and practical hurdles that preclude access to sensitive personal information. In this regard, existing approaches offer unsatisfying trade-offs: off-the-shelf LLMs trained via supervised fine-tuning (SFT) (Tan et al., 2024) and reinforcement learning from human feedback (RLHF) (Bai et al., 2022; Christiano et al., 2017) are easy to deploy but tend to collapse toward central tendencies in their training data, obscuring the preference heterogeneity (Feng et al., 2024; Kirk et al., 2024; Lee et al., 2024; Slocum et al., 2025); demographic conditioning methods require sensitive personal data and labor-intensive profile construction, yet still fail to guarantee distributional fidelity (Argyle et al., 2023; Castricato et al., 2025; Park et al., 2024; Bisbee et al., 2024).

In this work, we address this gap by introducing *preference reconstruction theory*, a framework that formalizes pop-

[1]Anonymous Institution, Anonymous City, Anonymous Region, Anonymous Country. Correspondence to: Anonymous Author <anon.email@domain.com>.

Preliminary work. Under review by the International Conference on Machine Learning (ICML). Do not distribute.

---

[1]See Appendix E for an extensive review on this topic.

ulation preference alignment as a representation learning problem, inspired by revealed preference theory in economics. The core insight is that population-level preferences can be reconstructed by appropriately weighting a functional basis of proxy agents. We focus specifically on social survey data as our alignment signal, as they are publicly available, transparently documented, and designed to represent well-defined populations, yet typically withholding sensitive demographic information that would enable individual-level profile matching[2]. We operationalize this theory via P2P, an inference-based alignment system with a modular two-stage design. Stage 1 uses prompt engineering and entropy-based adaptive sampling to construct a diverse pool of agents with different response patterns, aimed to span the latent preference space of a topic domain. Stage 2 employs L1-regularized regression to select a representative and compact ensemble of agents whose response distributions match those of the target human population. This modularity allows each stage to be adapted or extended independently, accommodating different sampling strategies or aggregation methods as needed. We validate the design through simulation studies and evaluate P2P on the American Trends Panel and World Values Survey, demonstrating robust performance across topics and locales at low cost.

Our contributions are as follows:

- **Problem formulation.** We formalize population preference alignment as a problem of representation learning, and introduce *preference reconstruction theory*, which states that learnable population preference can be represented using a functional basis of proxy agents with appropriately determined aggregation weights.

- **Modular alignment system.** We operationalize preference reconstruction theory via Prompts to Proxies (P2P), a two-stage system that combines entropy-based adaptive sampling for diverse agent generation with L1-regularized regression for compact ensemble selection, requiring no fine-tuning, no demographic data, and modest resources to run.

- **Empirical validation.** We evaluate P2P across 14 ATP waves and the World Values Survey, showing 43% improvement over prompting baselines, cross-locale generalization with theoretically grounded error analysis, and competitive fidelity against SFT approaches at less than 3% of the training data.

## 2. Modeling Preferences via Functional Bases

---

[2]We expect the theory to extend to proprietary data in the format of categorical responses.

**Alignment as representation learning** We formalize human-AI preference alignment as a representation learning problem, inspired by the revealed preference theory (Mas-Colell et al., 1995; Samuelson, 1938). Let $\mathcal{P}$ denote the latent human preference space. An *individual* human preference is represented as a vector in this space, $p \in \mathcal{P}$, which gets revealed via a decision-making mechanism $h$ into observed responses, denoted by vector $r \in \mathcal{R} \subset \mathbb{R}^n$. Similarly, let $\widehat{\mathcal{P}}$ be the latent model preference space.[3] A model preference is represented as $\hat{p} \in \widehat{\mathcal{P}}$, which manifests through the inference mechanism $f$ into the same observed response space $\mathcal{R}$. The representation learning process can thus be abstracted as:

$$p \xrightarrow{h} r \xrightarrow{f^{-1}} \hat{p} \,, \tag{1}$$

where $f^{-1}$ is implemented via a learning algorithm[4]. We introduce learnability of preference in a revealed sense.

**Definition 2.1.** Individual human preference is *learnable in the revealed sense* if $h(\mathcal{P}) \subset \mathrm{conv}(f(\widehat{\mathcal{P}}))$.

The definition using convex hull accommodates cases where individual human preference is not learnable by a single model but can be learned using an ensemble.

**Learning population preference via survey data** While the above describes the learning process of an individual preference, alignment of large language models (LLMs) is usually done using responses from a human population. However, this target population is typically ill-defined for general-purpose LLMs, which have been through distinct stages of pretraining, supervised finetuning (SFT) and reinforcement learning from human feedback (RLHF), with data sources from different user groups. In this paper, we focus on preference alignment using aggregate responses in general social surveys, because these surveys are readily available, transparently documented, and designed to represent the views of well-defined target population. Formally, aggregate responses in a survey are defined as weighted averages of individual responses:

$$r_{\mathrm{pop}} = \sum_{i=1}^{I} w_i h(p_i) \,, \tag{2}$$

where $w_i$ is the weight of individuals with preference $p_i$ in the population, $I$. Accordingly, we extend our learnability definition to cover human population preference.

**Definition 2.2.** Human population preference is *learnable in the revealed sense* if $\mathrm{conv}\left(h(\mathcal{P})\right) \subset \mathrm{conv}(f(\widehat{\mathcal{P}}))$.

---

[3]We intentionally define the model preference space $\widehat{\mathcal{P}}$ as an abstract space to accommodate the fact that both training and prompting can lead to different model preferences. In this paper, we use model and LLM interchangeably, as we contextualize alignment in the NLP context.

[4]Finding $\hat{p}$ such that $f(\hat{p}) \approx r$, not a literal inverse.

It follows readily that human population preference is learnable in the revealed sense if and only if individual human preference is learnable in the revealed sense (cf. Proposition A.4). Hence, in this paper we focus the discussion on population preference, with the expectation that the results extend to individual cases.

Methodologically, a population preference, revealed through aggregate responses $r_{\text{pop}}$, can be learned using two different paradigms. The first hinges on the stronger assumption that $\text{conv}(f(\hat{\mathcal{P}})) = f(\hat{\mathcal{P}})$, and trains a single model, called a *representative LLM agent*, which has a preference $\hat{p}_{\text{pop}} \in \widehat{\mathcal{P}}$ such that $r_{\text{pop}} = f(\hat{p}_{\text{pop}})$[5]. The second, which relates to our work, relies on an ensemble of $J$ models, whose preferences $\{\hat{p}_j\}_{j=1}^J$ satisfy $r_{\text{pop}} = \sum_{j=1}^J \tilde{w}_j f(\hat{p}_j)$, with $\{\tilde{w}_j\}_{j=1}^J$ convex weights. Under this paradigm, a well-explored method is *demographic conditioning* (Argyle et al., 2023; Castricato et al., 2025; Park et al., 2024), which aims to identify the one-to-one map between $p_i$ and $\hat{p}_i$ through restrictive access to sensitive personal information and labor-heavy profile matching. Here we provide three observations in relation to the two paradigms and offer short proofs for each in Appendix A.

**Observation 2.3.** *If human population preference can be learned by a representative LLM agent, it can also be learned using a non-trivial[6] ensemble of LLM agents.*

**Observation 2.4.** *If human population preference is learnable in the revealed sense, there could be cases where it can be learned by an ensemble but not by demographic conditioning nor a representative agent.*

**Observation 2.5.** *If human population preference can learned by demographic conditioning, the ensemble that can learn it is not unique.*

**Preference reconstruction theory**   Inspired by Observations 2.3-2.5, we propose *preference reconstruction theory*: instead of trying to find a one-to-one map between $p_i$ and $\hat{p}_i$, we can construct a *functional basis*—a set of proxy LLM agents with distinct preferences $\{\hat{p}_j\}_{j=1}^J$ spanning the relevant dimensions of human preferences—and then recover the human population preference through appropriate selection and weighting of these agents based on observational data. Figure 1 provides a functional perspective on preference reconstruction theory, together with a geometric interpretation of when the learnability assumption is violated (*inadequate basis*).

**Alignment as a two-stage problem**   Preference reconstruction via a functional basis can be formulated as a con-

---

[5]Appendix A formalizes this and connects to the existence of representative agent in economics.

[6]A representative agent is by definition a degenerate ensemble.

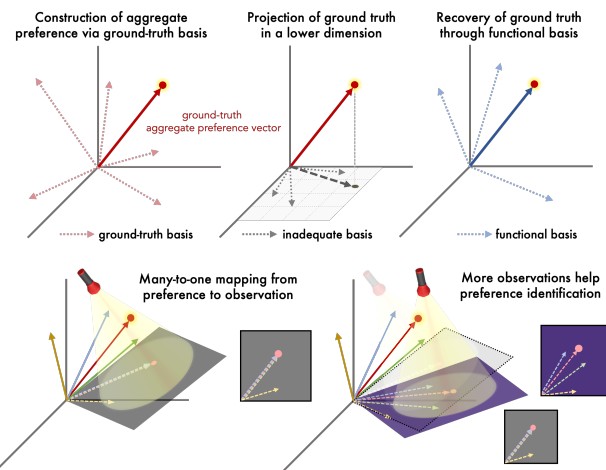

*Figure 1.* **Top**: Finding a functional basis to recover latent ground-truth preference. **Bottom**: Identification of latent preference through multiple observations.

crete learning problem:

$$\min_{\{\hat{p}_j, \tilde{w}_j\}_{j=1}^J} L\left(r_{\text{pop}}, \sum_{j=1}^J \tilde{w}_j f(x_0; \hat{p}_j)\right), \qquad (3)$$

where $L$ is a loss function.

Directly optimizing (3) is challenging, especially as the dimension $J$ is an optimizer per se.

As a solution strategy, we treat alignment as a two-stage problem with distinct stage goals. The first stage prioritizes the construction of a diverse set of proxy agents, measured in a revealed sense by their response variability. This goal aligns with the notion of *pluralistic alignment* (Sorensen et al., 2024b), which stresses the importance of representing a spectrum of human values rather than converging on a single ideal. The second stage concentrates on selecting a parsimonious ensemble from the set of candidate agents, based on the fit between ensemble aggregate responses and human ground truth. To avoid overfitting and test whether the ensemble has indeed learned the population preference, we partition survey questions into training and test sets, using the training set for selection and test set for validation. This solution strategy allows us to implement preference reconstruction modularly.

## 3. Prompts to Proxies (P2P)

To operationalize our preference reconstruction strategy in the survey context, we engineer a modular alignment system called **Prompts to Proxies (P2P)** (Figure 2). The system consists of two core components: (1) active endowment generation, powered by a dynamic attribute bank, and (2) regression-based aggregation, where constrained a variable selection algorithm forms the backbone. They concretize the

two stages of alignment introduced in the previous section and enable the generation of a compact set of opinionated LLM agents to emulate population-level survey results.

### 3.1. Survey Splitting and Evaluation Protocol

To support the alignment procedure, the full set of survey questions denoted $\mathcal{Q}$ are partitioned into training, validation, and test subsets $\{\mathcal{Q}_i\}_{i=\text{train,valid,test}}$. $\mathcal{Q}_{\text{train}}$ is used in Stage 1 while $\mathcal{Q}_{\text{train}}$ and $\mathcal{Q}_{\text{valid}}$ are used in Stage 2. $\mathcal{Q}_{\text{test}}$ serves as the validation to test preference generalizability.

### 3.2. Attributes as Control Handles

Language models, as next-token predictors, lack intrinsic preferences like humans. Their outputs are governed by statistical associations learned from text corpora and shaped by the user prompts. The prompt space is far too vast to explore in an unstructured manner. To address this, we adopt a structured prompting strategy based on *attributes*—interpretable factors, such as ideology, dispositions, or values, whose variation is likely to influence preference expression. P2P treats attributes as *control handles* for steering model behavior.[7]

P2P constructs endowments (persona profiles) by drawing attributes from the *attribute bank* and instantiating each attribute with a specific value through a specialized LLM termed *endowment model* (cf. Appendix B.2.2)[8]. The preset attribute bank for P2P lists example attributes taken from different social science disciplines, such as economics and political science. The bank is organized hierarchically: individual attributes are grouped into **modes**, each representing a coherent subject area (e.g., cognitive biases, political ideology). These modes are further clustered into higher-level **templates**, which reflect different design purposes or theoretical orientations. Figure 4 offers a structured view of the attribute bank and the endowment generation workflow.

While the attribute bank ensures structured coverage, we also enable dynamic discovery through two *freeform templates* that are specific to the survey under analysis. The **survey** template prompts an LLM *attribute learner* to extract decision-making factors from the overall survey topic. The **question** template derives attributes for a specific survey question. These attributes are added to the bank only for the survey run. This hybrid approach balances theory- and data-driven methods, enabling human-AI co-discovery of key preference descriptors.

---

[7]This contrasts from the human case, where such attributes are typically viewed as reflections of latent preferences. P2P's active endowment generation deliberately leverages this inversion: rather than inferring attributes from preferences, it uses attributes to shape the model's expressed preferences.

[8]The use of endowment model ensures the created endowment remains coherent and interpretable to human readers.

### 3.3. Stage 1: Active Endowment Generation

Stage 1 of P2P constructs a diverse preference basis without exhaustively enumerating attribute combinations. It begins by invoking an attribute learner to infer survey-specific attributes from the training questions, augmenting the attribute bank. Endowment generation then proceeds via adaptive sampling by mode: After initializing with equal endowments per mode, subsequent steps follow a bandit-style strategy—prioritizing modes that yield high response variability while exploring new modes for under-represented question items (see full Algorithm 1 in Appendix B.3).

**Tracking response diversity with variability score**  To measure preference diversity of agents in a revealed sense, we introduce the notion of **question entropy**:

$$H_i(\mathcal{A}) = \frac{-\sum_{k=1}^{K_i} p_{ik} \log_2 p_{ik}}{\log_2 K_i} \, , \qquad (4)$$

where $H_i(\mathcal{A})$ denotes the normalized entropy of question $i$ based on responses from a group of agents $\mathcal{A}$, $K_i$ is the number of unique response options $\{1, \ldots, K_i\}$ for question $i$, and $p_{ik}$ is the empirical proportion of responses selecting option $k$. The denominator normalizes for scale, ensuring that entropy is comparable across questions with different numbers of choices. A high question entropy indicates greater response diversity for that specific question. By restricting the group $\mathcal{A}$ to agents sampled from a particular mode (subject area), we can compute question entropy at the mode level, termed **mode-question entropy**, and compare the response variability across modes.

Adaptive sampling references the **variability score** for each mode. Formally, it is defined as

$$V(\mathcal{A}_{\text{mode}}) = \sum_{i \in \mathcal{Q}_{\text{train}}} \frac{H_i(\mathcal{A}_{\text{mode}})}{H_i(\mathcal{A}_{\text{total}})} \, , \qquad (5)$$

where $H_i(\mathcal{A}_{\text{mode}})$ denotes the entropy of responses to question $i$ based on the cumulative set of agents from a given mode up to the current generation step, and $H_i(\mathcal{A}_{\text{total}})$ is the corresponding question entropy across all agents generated so far. Computed this way, the variability score favors modes that are capable of generating endowments which elicit diverse responses to questions that other modes typically fail to diversify. In P2P, we task a dedicated class object, Tracker, to compute the variability score for each mode after each generation round (see Algorithm 2 in Appendix B.3). The scores are then passed to a softmax function to produce a probabilty distribution over modes, guiding endowment sampling in the next generation step via a multimonial draw—balancing exploitation of top-performing modes with exploration of under-performing ones.

**Patching low-entropy questions**  In addition to the existing trade-off between exploitation and exploration, we

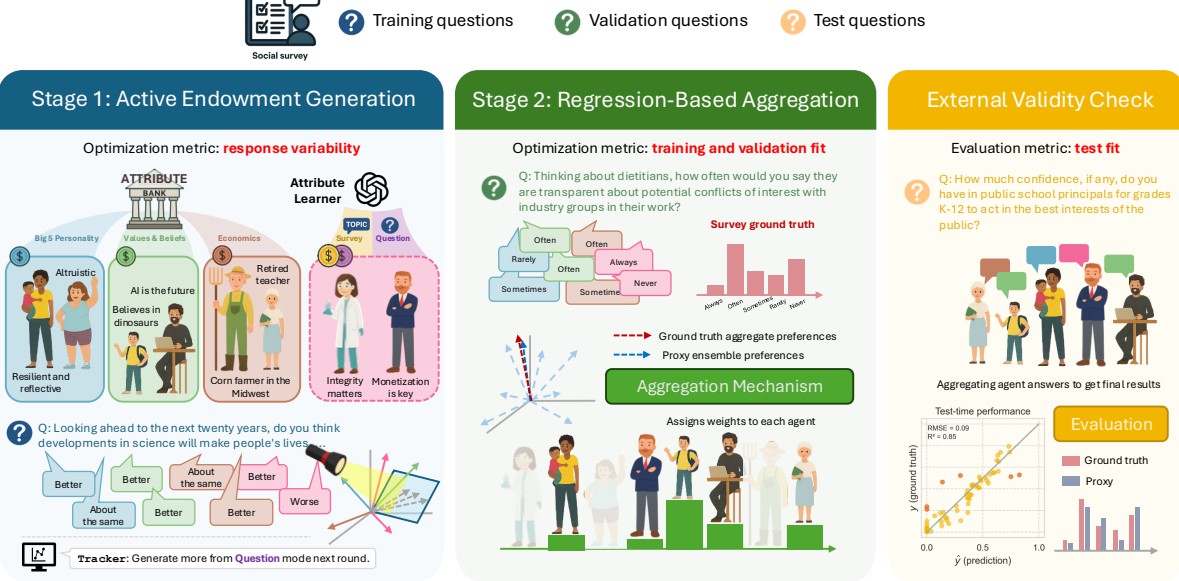

*Figure 2.* **System Overview of the Prompts to Proxies (P2P) Framework.** P2P operationalizes preference alignment as a two-stage process inspired by revealed preference theory. In **Stage 1**, active endowment generation uses a structured attribute bank and entropy-guided adaptive sampling strategies to construct diverse agent personas, called endowments. In **Stage 2**, regression-based aggregation assigns weights to reconstruct population-level preference patterns. An external validity check accesses how well the weighted ensemble predicts aggregate responses on held-out questions, assessing out-of-sample generalization.

adopt an additional patching strategy, which we call *question patching*. At the end of each update step, as the `Tracker` computes the variability scores for each existing mode, it also keeps track of the questions with the lowest entropy values. When question patching is activated, the attribute learner will be called in the next generation step to exclusively infer attributes on each of the lowest-$k$ entropy questions, where $k$ is user-defined. The split of the endowment budget between active generation and patching generation is governed by a user-specified ratio.

To avoid wasting resources on persistently low-entropy and to promote cross-pollination between modes, we additionally introduce a *mixed mode* strategy. It is triggered after a question appears $t$ times in the lowest-$k$ entropy list, where $t$ is a user-defined threshold. When in effect, the strategy relocates the assigned endowment budge to a new mixed mode, which combines attributes from the question with those of the current top-performing mode, allowing diversity injection from high-performing modes while maintaining a question-specific focus. Figure 10 shows example endowments generated from a real run of P2P, illustrating how different templates and modes yield distinct profiles.

The adaptive sampling cycle continues until the generation budget $N_A$ (an upper bound on the maximum number of agents allowed) is exhausted. By iteratively targeting low-diversity questions, expanding the attribute space and promoting cross-pollination between modes, active endow-

ment generation progressively increases response diversity while preserving coverage over previously explored regions of the preference space (cf. Figure 5 in Appendix B.4).

### 3.4. Stage 2: Regression-Based Aggregation

With the proxy basis assembled, P2P enters the second stage: regression-based aggregation. The goal at this stage is to find a functional basis from the proxy basis to reconstruct the population preference based on the observational data. Depending on the endowment budget $N_A$ and the total number of trainable questions $\mathcal{Q}_{\text{train}} \cup Q_{\text{valid}}$, we could end up in a situation where there are more variables (proxy agents) than observations (trainval questions). Consequently, a variable selection algorithm is vital. Besides, even with $|\mathcal{Q}_{\text{train}} \cup Q_{\text{valid}}| >> N_A$, variable selection allows us to retain a parsimonious agent ensemble, saving inference costs on further simulations. We consider regression methods with an added L1 penalization term and use cross-validation (CV) for hyperparameter tuning. The objective of the learning problem can be stated as

$$\min_{\mathbf{w}} \sum_{i \in \mathcal{Q}_{\text{train}}} \sum_{k=1}^{K_i} \left( r_{ik} - \sum_{j=1}^{N_A} w_j d_{j,ik} \right)^2 + \alpha \Omega(\mathbf{w}) , \quad (6)$$

$$\text{s.t.} \quad \sum_{k=1}^{K_i} \sum_{j=1}^{N_A} w_j d_{j,ik} = 1 \quad \forall i \in \mathcal{Q}_{\text{train}} , \quad (7)$$

where $\mathbf{w} \in \mathbb{R}^{N_A}$ is the weight vector for the agents, $r_{ik}$ denotes the ground-truth proportion of responses selecting option $k$ for question $i$, $d_{j,ik}$ is a dummy response variable equal to 1 if agent $j$ selects option $k$ and 0 otherwise, and $\alpha$ is the hyperparameter for the penalization. When $\Omega$ consists of only a L1 norm, the formulation amounts to constrained lasso. When $\Omega$ subsumes an additional L2 norm, it becomes constrained elastic net.

To validate L1-regularized regression as a credible aggregation algorithm, we conduct extensive simulation studies for constrained lasso (Appendix C), where ground truth population preference is constructed using synthetic agents[9].

Our simulation studies yield three main findings. First, constrained lasso faithfully recovers the ground-truth agents from a large candidate pool under an adequate number of observations, including minority profiles with low weights, and only resorts to reconstruction when observations are extremely scarce (Appendix C.1). Second, when ground truth agents are removed from the candidate pool, constrained lasso remains effective in selecting a proxy ensemble to recover ground truth even when the number of proxy agents is small (Appendix C.2). Third, proxy ensembles with insufficient response entropy fail to recover ground-truth preferences, while those with excess entropy show only marginal deterioration, corroborating our choice of response variability as the optimization goal of Stage 1 of P2P (Appendix C.3). Together, these results provide constructive evidence for P2P's design choices.

# 4. Empirical Evaluation

Having validated P2P's design choices through simulation, we now evaluate it on real human survey data. We use the American Trends Panel (ATP) (Pew Research Center), a nationally representative U.S. survey conducted by Pew Research Center. We obtain the digitalized versions of these datasets from the OpinionsQA benchmark (Santurkar et al., 2023), which repurposed ATP responses to study value pluralism and misalignment in language models. We assess P2P across 14 ATP waves, spanning diverse socio-economic topics. We then evaluate locale generalizability on the World Values Survey, conduct error analysis and ablation studies, and stress-test against a distribution-calibrated model under topic shift.

## 4.1. Experimental Setup

For each survey, we split the questions according to a 7 : 1.5 : 1.5 ratio into the training, validation, and test sets. To benchmark P2P, we create two prompting-based ensemble baselines with different endowment generation logic but

feed both through the same regression-based aggregation stage as P2P for proxy selection and test set evaluation. This allows the baselines to benefit from the same data-driven selection mechanism, isolating the contribution of P2P's endowment generation strategy.

**Vanilla baseline** Endowments are generated using an LLM prompted[10] directly to generate diverse agent profiles conditioned only on the survey topic.

**PERSONA** Endowments are sampled randomly[11] from the demographically conditioned PERSONA dataset (Castricato et al., 2025), which contains 1000 persona profiles derived from U.S. census data.

For all methods, we set the endowment generation budget to 300. For P2P, initial sampling generates 10 endowments for each mode; active endowment generation is run in 5 update steps with the preset attribute bank, question patching (lowest 3; 75% endowment budget), and 10 attributes drawn for each endowment generation. Unless otherwise noted, all models use Gemini-2.0-flash as the backend[12], and constrained lasso is used for Stage 2 selection.

## 4.2. Main Results: Baseline Comparison on ATP Waves

As Table 1 shows, P2P consistently outperforms both baselines, achieving an average test MSE of 0.014 compared to 0.026 (Vanilla) and 0.029 (PERSONA)—a 43% improvement over the best baseline. For P2P, 12 out of 14 waves achieve mean test MSE (across 3 runs) below 0.02, confirming generalization across topics without wave-specific tuning. The agent pool created using P2P also achieves higher diversity than the baselines, indicated by the higher average entropy (0.53 vs. 0.43-0.45), corroborating our simulation finding that response diversity supports preference reconstruction (Appendix C.3). Notably, this diversity gain is cost-efficient, as P2P runs at approximately $0.8 per survey, similar to Vanilla ($0.7) and substantially cheaper than PERSONA ($1.3, excluding persona generation overhead)[13]. Performance comparison is robust under other distributional metrics (1-JSD, EMD) (Appendix D).

## 4.3. Locale Generalization: World Values Survey

To probe whether P2P generalizes beyond U.S. populations and ATP-specific survey design, we evaluate it on

---

[9]This allows us to create a clean environment where alignment result is not affected by the learnability condition (Def. 2.2).

[10]The endowment generation prompts for P2P and the Vanilla baseline can be found in Appendix B.2.

[11]We use random sampling as it preserves the representativeness of the dataset. In Appendix D.1.1, we provide baseline variants using K-means clustering as a pre-processing step. Clustering does not improve performance but increases cost.

[12]See Appendix D.1.2 for raw model performance.

[13]The PERSONA pipeline (Castricato et al., 2025) requires resampling and re-inference, adding further API usage costs.

*Table 1.* Performance comparison across 14 ATP waves. Test MSE, entropy, and cost (USD) reported. `P2P` reports mean over 3 runs. Baselines report single runs, as repeated runs on W27, W32, W42, and W45 show negligible variance (MSE std < 0.005), confirming stability (Appendix D). Δ shows relative test MSE improvement over the best baseline.

| Wave | Topic | # Qs | Vanilla MSE | Vanilla Ent. | Vanilla Cost | PERSONA MSE | PERSONA Ent. | PERSONA Cost* | P2P MSE | P2P Ent. | P2P Cost | Δ |
|------|-------|------|-----|------|------|-----|------|-------|-----|------|------|-----|
| W26 | Guns | 78 | .015 | .50 | $0.5 | .021 | .37 | $1.0 | **.010** | .55 | $0.6 | 33% |
| W27 | Auto | 96 | .030 | .40 | $0.6 | .042 | .40 | $1.3 | **.022** | .54 | $0.7 | 27% |
| W29 | Gender | 77 | .030 | .45 | $0.5 | .030 | .44 | $1.0 | **.016** | .51 | $0.6 | 47% |
| W32 | Comm. | 98 | .020 | .55 | $0.6 | .026 | .56 | $1.3 | **.016** | .62 | $0.7 | 20% |
| W34 | Biomed | 67 | .022 | .43 | $0.5 | .020 | .39 | $0.9 | **.011** | .54 | $0.5 | 45% |
| W36 | Lead. | 139 | .033 | .34 | $0.9 | .035 | .39 | $1.8 | **.019** | .48 | $1.0 | 42% |
| W41 | 2050 | 90 | .022 | .45 | $0.6 | .023 | .48 | $1.2 | **.013** | .55 | $0.8 | 41% |
| W42 | Science | 129 | .034 | .32 | $0.9 | .038 | .29 | $1.7 | **.011** | .43 | $1.0 | 68% |
| W45 | Misinfo | 95 | .016 | .39 | $0.6 | .028 | .36 | $1.2 | **.008** | .44 | $0.7 | 50% |
| W49 | Privacy | 98 | .045 | .41 | $0.6 | .047 | .41 | $1.3 | **.021** | .53 | $0.7 | 53% |
| W50 | Family | 128 | .027 | .36 | $0.8 | .024 | .36 | $1.7 | **.013** | .44 | $0.9 | 46% |
| W54 | Econ | 116 | .022 | .50 | $0.8 | .026 | .54 | $1.5 | **.014** | .57 | $0.9 | 36% |
| W82 | Global | 104 | .025 | .52 | $0.7 | .019 | .48 | $1.4 | **.008** | .57 | $0.8 | 58% |
| W92 | Poli. | 77 | .025 | .59 | $0.5 | .031 | .57 | $1.0 | **.017** | .61 | $0.6 | 32% |
| Average | | | .026 | .45 | $0.7 | .029 | .43 | $1.3 | **.014** | .53 | $0.8 | 43% |

*PERSONA cost only covers the response elicitation stage—higher if persona generation tokens are included.

*Table 2.* `P2P` performance on WVS Wave 7 across three locales. Values are mean ± std over 3 runs. (Full details in Appendix D.)

| Metric | US | GB | HK |
|--------|-----|-----|-----|
| Entropy ↑ | .572 ± .015 | .571 ± .004 | .548 ± .021 |
| Test MSE ↓ | .015 ± .001 | .014 ± .001 | .024 ± .001 |
| Cost ↓ | 1.76 ± .01 | 1.71 ± .00 | 1.74 ± .01 |

World Values Survey (WVS) Wave 7 (Haerpfer et al., 2022), using cleaned data from Cao et al. (2025). We focus on three representative regions: the United States, Great Britain, and Hong Kong SAR. Under the same experimental protocol[14], `P2P` achieves comparable performance on US and GB (MSE below 0.015), with somewhat weaker results on HK (MSE around 0.024) (Table 2).

### 4.4. Error Analysis: Transfer vs. Native Alignment

One plausible explanation for the WVS performance gap is that LLMs trained primarily on English and Western-centric data better reflect Western populations (Durmus et al., 2024). Yet, we want to probe it further by connecting to our theoretical framework: How much of the gap is explained by insufficient agent diversity and can we characterize the learnability limits? To address these questions, we take US endowments, which have higher entropy, and apply them to HK via `P2P` Stage 2 by refitting regression weights. We call this HK-transfer and compare it against those in Table 2 (called US-native and HK-native).

[14]We pass the locale information to the endowment model to ensure regional relevance of the generated endowments.

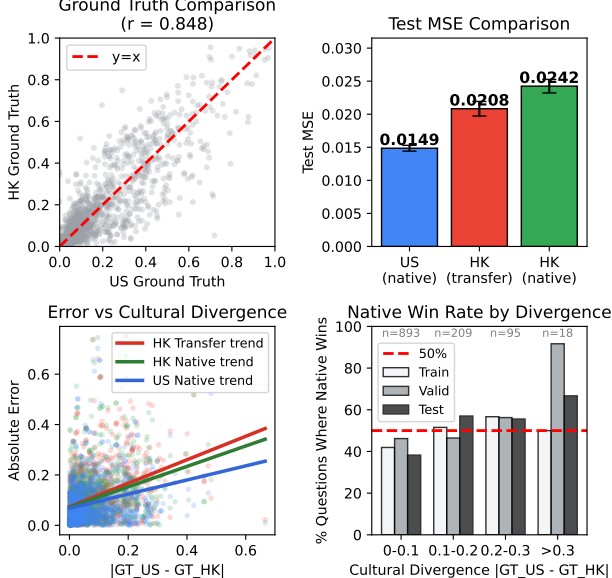

*Figure 3.* **Error Analysis of Hong Kong WVS Results.** We compare UK-native and HK-native with HK-transfer (US endowments applied to HK via `P2P` Stage 2). **(A)** Ground truth divergence between populations ($r = 0.848$). **(B)** Transfer outperforms native in overall performance, corroborating the functional basis perspective. **(C)** Error gap scales with cultural divergence, suggesting learnability limits (Definition 2.2). **(D)** Despite higher test MSE, HK-native outperforms transfer on culturally divergent questions, indicating that localization helps where cultures genuinely differ.

Figure 3 presents four complementary analyses. We define cultural divergence for each question-option pair as $|\text{GT}_{US} - \text{GT}_{HK}|$, the absolute difference in response proportions between populations. Panel (A) shows correlated

but distinct US and HK ground-truth response patterns ($r = 0.848$), with variation around the diagonal confirming cultural differences. Panel (B) shows that HK-transfer achieves *lower* test MSE than HK-native, corroborating the functional basis perspective—a more diverse agent pool facilitates preference reconstruction, even when generated for a different locale[15]. However, Panels (C) and (D) reveal important nuances. Panel (C) shows that prediction error scales with cultural divergence, offering an empirical handle on learnability limits (Definition 2.2). In culturally distinct dimensions, Gemini-based ensembles exhibit a clear bias toward US ground truth. As indicated by the trends, conditioning on locale information provides a mild mitigation. Panel (D) further corroborates this, showing that despite higher test MSE, HK-native outperforms HK-transfer on culturally divergent questions. The takeaway is that while cross-locale transfer can leverage diversity on average, locale-specific generation may capture culturally distinctive preferences that a foreign basis cannot fully reconstruct. The analyses also demonstrate P2P's potential as a benchmark for model steerability and bias research (cf. Appendix F).

### 4.5. Ablation Studies

We conduct ablation studies on ATP Wave 42 to isolate component contributions. Full results are in Appendix D.3.

**Attributes**  Disabling question patching reduces entropy from 0.445 to 0.399 and increases test MSE from 0.010 to 0.011, while disabling both freeform templates (question & survey) further degrades MSE to 0.013. This suggests that human-AI co-discovery of preference descriptors yields meaningful performance gains. Varying the per-endowment attribute cap (10 vs. 20) has minimal effect.

**Endowment budget**  Performance improves with budget (110-450 agents) but plateaus after 3 update steps (210 agents). A no-update baseline (110 agents) underperforms, validating the benefit of adaptive sampling.

**Model backend**  We test P2P's performance on five model backends: GPT-4.1-mini, GPT-4.1-nano, Gemini-2.0-flash, Gemini-2.5-flash and a locally hosted Qwen. GPT-4.1-mini achieves the best entropy (0.471) and test MSE (0.009). Gemini-2.0-flash ranks second at one-quarter the cost. Smaller models (GPT-4.1-nano and Gemini-2.5-flash-lite) perform worse, consistent with our intuition that limited model capacity constrains the learnability (Definition 2.2).

**Regression-based aggregation**  Replacing regression with simple averaging of agent responses increases test MSE

---

[15]This result cuts against demographic conditioning approaches, which assume locale-matched profiles are necessary for faithful preference reconstruction.

from 0.0104 to 0.0254, confirming that regression is essential for learning population preferences, not just ensemble compression. Switching loss weighting from option level to question level has minimal effect on W42 results.

### 4.6. Stress Test: Comparison with an SFT-Aligned Model under Topic Shift

P2P is designed as an inference-based alignment method for limited-data and limited-compute regimes, so our baselines focus on prompting-based methods under similar constraints. However, supervised finetuning (SFT) offers an alternative when more data is available. For instance, Cao et al. (2025) train a distribution-calibrated model on pooled WVS data from 46 countries (approx. 8,400 training entries) to predict answer distributions conditioned on locale.

We stress-test P2P on their protocol, which induces a topic shift: training on general attitudinal and religious-ethical items (Q1+Q2) and testing on political-cultural items (Q3). We apply P2P to three locales (the United States, Great Britain, and Hong Kong SAR), using only 172-185 local training entries and no finetuning. Despite using less than 3% of the training data and no SFT, P2P achieves comparable Jensen-Shannon Divergence (1-JSD) (0.71-0.75 vs. 0.78), with matching or better majority-class prediction accuracy (MCPA) (0.47-0.64 vs. 0.43). We interpret this as evidence that some preference structure transfers across topics though we do not assume such transfer is guaranteed. Studying cross-topic generalization remains an important direction for future work (Appendix F). Details on the stress test are presented in Appendix D.4.

## 5. Conclusion

This work formalizes preference alignment as a problem of preference reconstruction and operationalizes it as a two-stage process: first, constructing a functional basis of proxy agents; and then, recovering aggregation weights through supervised learning. We introduce P2P, a modular alignment system that implements preference reconstruction, combining entropy-based adaptive sampling with regression-based aggregation. Our simulation results corroborate P2P's design choices. Empirical performance shows P2P can recover preference patterns observed in real human survey data while remaining competitive against SFT-aligned models despite using significantly less data. Future work can extend P2P to accommodate freeform textual output and aggregation, analyze its use as a benchmark for model steerability, devise re-anchoring strategies for non-stationary preference, and incorporate semantic and psychometric techniques for improved robustness and generalization (Appendix F). Our work offers a first step towards these endeavors and a practical platform for interdisciplinary research.

## Impact Statement

This work uses publicly available, anonymized survey datasets with sensitive identifiable information removed. While the large language model (LLM) backends have been trained to align with human values, they may still propagate social biases and stereotypes, and some generated endowment profiles may contain opinionated content. Our goal is methodological—studying preference reconstruction at the aggregate level rather than individual personalization—which reduces profiling risks. We see potential for positive impact on pluralistic alignment and principled use of AI in social science research, but stress that the agent ensemble is an auxiliary tool for understanding human choice patterns and should not replace humans in critical policy-making scenarios.

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

## A. Modeling Preferences via Functional Bases: Proofs

**Proposition A.0** (Existence of a representative agent). *Let $\widetilde{\mathcal{P}}$ be a latent preference space (regardless of human or model), with an individual preference denoted by $\tilde{p} \in \widetilde{\mathcal{P}}$. Let $g$ be the map from $\widetilde{\mathcal{P}}$ to the observed response space $\widetilde{\mathcal{R}} \subset \mathbb{R}^n$. Then the following two statements are equivalent:*

1. *The image $g(\widetilde{\mathcal{P}}) \subset \widetilde{\mathcal{R}}$ is convex.*

2. *For any aggregate response $\tilde{r}_{\mathrm{pop}} = \sum_{i=1}^{I} w_i \tilde{r}_i$ with $\{w_i\}_{i=1}^{I}$ convex weights and $\tilde{r}_i = g(\tilde{p}_i)$, there exists $\tilde{p}_{\mathrm{pop}} \in \widetilde{\mathcal{P}}$ such that $g(\tilde{p}_{\mathrm{pop}}) = \tilde{r}_{\mathrm{pop}}$.*

*Proof.* First, we prove $(1) \implies (2)$.

By definition, $\tilde{r}_i = g(\tilde{p}_i) \in g(\widetilde{\mathcal{P}})$ for all $i \in \{1, \ldots, I\}$. Since $\{w_i\}_{i=1}^{I}$ are convex weights ($\sum_{i=1}^{I} w_i = 1$ and $w_i \geq 0$ for all $i \in \{1, \ldots, I\}$), and $g(\widetilde{\mathcal{P}})$ is convex, it follows that

$$\sum_{i=1}^{I} w_i \tilde{r}_i \in g(\widetilde{\mathcal{P}}).$$

By the definition of image, it follows that there exists $\tilde{p}_{\mathrm{pop}} \in \widetilde{\mathcal{P}}$ such that

$$g(\tilde{p}_{\mathrm{pop}}) = \sum_{i}^{I} w_i \tilde{r}_i \triangleq \tilde{r}_{\mathrm{pop}}.$$

Claim: $(2) \implies (1)$.

Let $\{\tilde{r}_i\}_{i=1}^{I}$ denote any finite collection of observed responses with $\tilde{r}_i \in g(\widetilde{\mathcal{P}})$, and $\{\lambda_i\}_{i=1}^{I}$ be any convex weights. Consider $\tilde{r}_{\mathrm{agg}} = \sum_{i=1}^{I} \lambda_i \tilde{r}_i$. By (2), there exists $\tilde{p}_{\mathrm{agg}} \in \widetilde{\mathcal{P}}$ such that

$$\tilde{r}_{\mathrm{agg}} = g(\tilde{p}_{\mathrm{agg}}) \in g(\widetilde{\mathcal{P}}). \tag{8}$$

Therefore, $g(\widetilde{\mathcal{P}})$ is convex by definition. $\square$

*Remark* A.1 (Sufficient condition for representative agent). If the decision-making mechanism $g$ maps a preference into a convex subset of a probability simplex $\Delta^{n-1}$, then any aggregate of individual responses is realizable by a representative agent.

*Proof.* A convex subset of a probability simplex is by definition convex. The result follows from the theorem. $\square$

The implications of this lemma are two-fold:

1. In human decision-making, the condition that $g(\widetilde{\mathcal{P}})$ is convex is less likely to be true unless we assume stochastic choice models with stringent functional forms (cf. Kirman, 1992; Jackson & Yariv, 2019).

2. In LLM decision-making, the condition that $g(\widetilde{\mathcal{P}})$ is convex is more likely to hold is more likely to be hold assuming that the model has considerable steerability. (E.g., through finetuning or prompting, we can effectively change the latent model preference embeddings and influence the output probability distributions.)

**Definition 2.1.** Individual human preference is *learnable in the revealed sense* if $h(\mathcal{P}) \subset \mathrm{conv}(f(\widehat{\mathcal{P}}))$.

**Definition A.2.** Human preference is *strongly learnable in the revealed sense* if $h(\mathcal{P}) \subset f(\hat{\mathcal{P}})$.

**Definition 2.2.** Human population preference is *learnable in the revealed sense* if $\mathrm{conv}\,(h(\mathcal{P})) \subset \mathrm{conv}(f(\widehat{\mathcal{P}}))$.

**Definition A.3.** Human population preference is *strongly learnable in the revealed sense* if $\mathrm{conv}\,(h(\mathcal{P})) \subset f(\hat{\mathcal{P}})$.

**Proposition A.4.** *Human population preference is learnable in the revealed sense if and only if individual human preference is learnable in the revealed sense.*

*Proof.* The "only if" part follows directly from $h(\mathcal{P}) \subset \mathrm{conv}(h(\mathcal{P}))$. Here we focus on the "if" part.

By definition, a human population preference is defined as

$$r_{\mathrm{pop}} = \sum_{i=1}^{I} w_i h(p_i).$$

Since individual human preference is learnable in a revealed sense, i.e., $h(\mathcal{P}) \subset \mathrm{conv}(f(\widehat{\mathcal{P}}))$, for any $p_i$, there exists $\{w_{ik}, \hat{p}_{ik}\}_{k=1}^{K_i}$ such that

$$h(p_i) = \sum_{k=1}^{K_i} w_{ik} f(\hat{p}_{ik}).$$

Plug it back into the definition of human population preference and we get

$$r_{\mathrm{pop}} = \sum_{i=1}^{I} w_i h(p_i) = \sum_{i=1}^{I} w_i \sum_{k=1}^{K_i} w_{ik} f(\hat{p}_{ik}) = \sum_{i=1}^{I} \sum_{k=1}^{K_i} w_i w_{ik} f(\hat{p}_{ik}) := \sum_{\tilde{i}=1}^{\tilde{I}} w_{\tilde{i}} f(\hat{p}_{\tilde{i}}).$$

The last step follows from re-indexing. Therefore, $\mathrm{conv}(h(\mathcal{P})) \subset \mathrm{conv}(f(\widehat{\mathcal{P}}))$. $\quad\square$

**Observation 2.3.** *If human population preference can be learned by a representative LLM agent, it can also be learned using a non-trivial[16] ensemble of LLM agents.*

*Proof.* By Proposition A.0, if human population preference can be learned by a representative LLM agent, then

$$\mathrm{conv}(h(\mathcal{P})) \subset \mathrm{conv}(f(\widehat{\mathcal{P}})) = f(\widehat{\mathcal{P}}).$$

For any $r_{\mathrm{pop}} \in \mathrm{conv}(h(\mathcal{P}))$, it follows from the definition of human population preference that

$$r_{\mathrm{pop}} = \sum_{i=1}^{I} w_i h(p_i) := \sum_{i=1}^{I} w_i r_i.$$

Since $r_i \in h(\mathcal{P}) \subset \mathrm{conv}(h(\mathcal{P})) \subset f(\widehat{\mathcal{P}})$, there exists $\hat{p}_i$ such that $f(\hat{p}_i) = r_i$.

It follows that

$$r_{\mathrm{pop}} = \sum_{i=1}^{I} w_i f(\hat{p}_i) \in \mathrm{span}\left\{f(\hat{p}_i)\right\}_{i=1}^{I}.$$

In words, we can construct an ensemble where each agent represents in preference a ground-truth individual in the human population (*demographic conditioning*).

NB: Here we implicitly assume that the human population consists of more than one individual with distinct preferences. If the population consists of only 1 individual, then the representative agent and the ensemble coincide. $\quad\square$

**Observation 2.4.** *If human population preference is learnable in the revealed sense, there could be cases where it can be learned by an ensemble but not by demographic conditioning nor a representative agent.*

*Proof.* Based on the proof of Observation 2.3 and Proposition A.0, we see that demographic conditioning hinges on the condition that $h(\mathcal{P}) \subset f(\widehat{\mathcal{P}})$, whereas representative agent relies on an even stronger condition that $\mathrm{conv}(h(\mathcal{P})) \subset f(\widehat{\mathcal{P}})$. If human population preference is learnable in the revealed sense, we only have the weaker condition[17]:

$$\mathrm{conv}(h(\mathcal{P})) \subset \mathrm{conv}(f(\widehat{\mathcal{P}})). \tag{9}$$

---

[16]A representative agent is by definition a degenerate ensemble.
[17]Equivalently, $h(\mathcal{P}) \subset \mathrm{conv}(f(\widehat{\mathcal{P}}))$ by Proposition A.4.

Consider the case where $h(\mathcal{P}) \not\subset f(\widehat{\mathcal{P}})$. I.e., there exists $p_i \in \mathcal{P}$ such that for *all* $\hat{p}_j \in \widehat{\mathcal{P}}$, $h(p_i) \neq f(\hat{p}_j)$. Then both demographic conditioning and representative agent fail. However, condition (9) implies that for any $r_{\mathrm{pop}} \in \mathrm{conv}\,(h(\mathcal{P}))$ there exists $\{\tilde{w}_j, \hat{p}_j\}_{j=1}^J$ such that

$$r_{\mathrm{pop}} = \sum_{j=1}^J \tilde{w}_j f(\hat{p}_j).$$

In other words, human population preference can be learned using an ensemble. $\square$

**Observation 2.5.** *If human population preference can learned by demographic conditioning, the ensemble that can learn it is not unique.*

*Proof.* If human population preference can be learned by demographic conditioning, we know from the proof of Observation 2.3 that $h(\mathcal{P}) \subset f(\widehat{\mathcal{P}})$ and that for any $r_{\mathrm{pop}} \in \mathrm{conv}(h(\mathcal{P}))$ there exist $\{\hat{p}_i\}_{i=1}^I$ such that

$$r_{\mathrm{pop}} = \sum_{i=1}^I w_i r_i = \sum_{i=1}^I w_i f(\hat{p}_i). \tag{10}$$

However, since $r_1 \in h(\mathcal{P}) \subset \mathrm{conv}(h(\mathcal{P}))$, we know there exists a finite set of $\{r_k^*\}_{k=1}^K$ with $r_k^* \in h(\mathcal{P})$ and convex weights $\{\lambda_k\}_{k=1}^K$ such that

$$r_1 = \sum_{k=1}^K \lambda_k r_k^*.$$

Plug this into (10) and we get

$$r_{\mathrm{pop}} = \sum_{k=1}^K w_1 \lambda_k r_k^* + \sum_{i=2}^I w_i r_i.$$

Since $r_k^* \in h(\mathcal{P}) \subset f(\widehat{\mathcal{P}})$, there exists $\hat{p}_k^*$ such that $f(\hat{p}_k^*) = r_k^*$.

In conclusion,

$$r_{\mathrm{pop}} \in \mathrm{span}\{f(\hat{p}_1^*), \ldots, f(\hat{p}_K^*), f(\hat{p}_2), \ldots, f(\hat{p}_I)\}.$$

$\square$

# B. Prompts to Proxies (`P2P`): Implementation Details

## B.1. Attribute Bank

For the preset attribute bank (Figure 4), we define three primary templates. The *core* template contains general demographic descriptors and serves as the default baseline. The *thematic* template includes topic-specific modes such as economics, politics, culture and history. The *theoretical* template groups attributes derived from canonical frameworks such as Maslow's hierarchy of needs (Maslow, 1943) and the Big Five personality traits (John & Srivastava, 1999). These templates and modes are not fixed: they serve as a structured starting point but are fully extensible. Users can define new templates, introduce domain-specific modes, or modify existing attribute sets to suit different experimental contexts. By sampling and varying attributes from this bank, we create prompts that instantiate proxy agents at different locations of the latent model preference space.

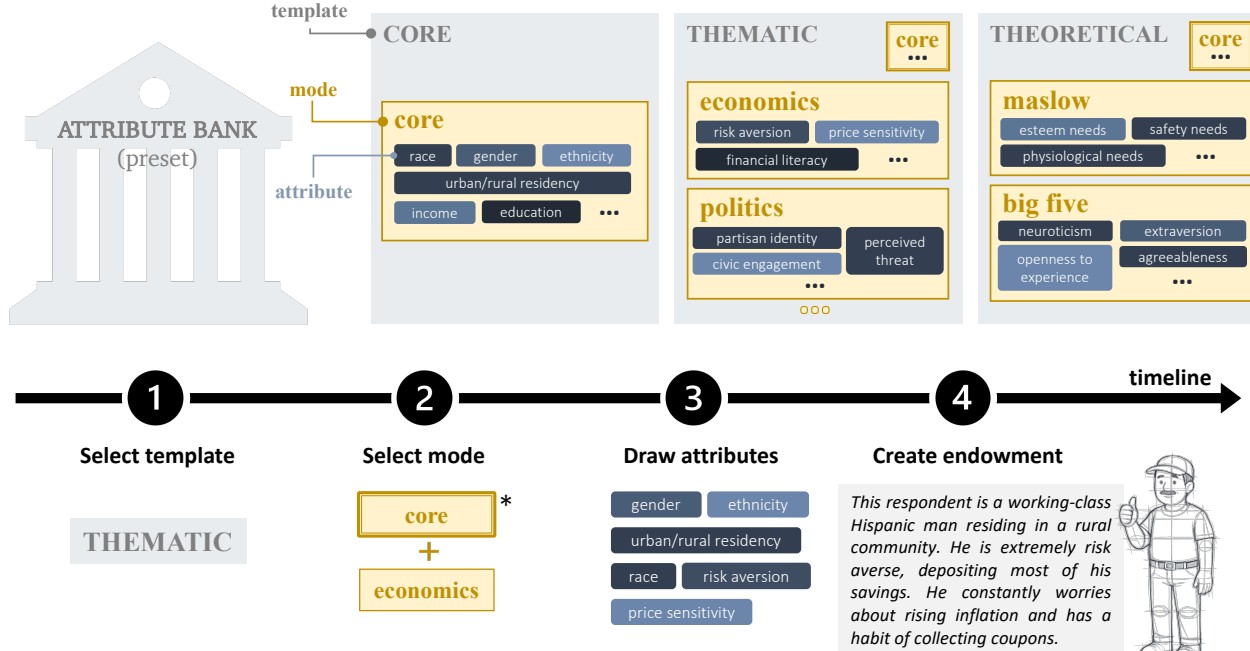

* Note: By design, thematic and theoretical templates inherit **core** demographic attributes to ground generated endowments.

*Figure 4.* **Attribute Bank and Endowment Generation Workflow.** The preset attribute bank is organized hierarchically by template (core, thematic, theoretical), mode (e.g., economics, politics, maslow, big five), and attribute (e.g., race, gender, ethnicity, urban/rural residency, risk aversion). Endowment is generated by a specialized LLM (`EndowmentModel`), which takes drawn attributes as input and outputs a profile with instantiation of these attributes.

## B.2. Model Prompts

### B.2.1. ATTRIBUTE LEARNER (ATTRIBUTELEARNER)

To generate attributes from a single question:

> Attribute Learner: Learning Attributes from A Single Question
>
> ```
> You are an intelligent research assistant trained to analyze individual
> survey questions and infer which human attributes might influence how
> different people respond.
> ```

```
You are given a single survey question.  Your task is to propose a list of
relevant human attributes|such as demographics, beliefs, values, personality
traits, or ideological leanings|that are likely to shape responses to this
question.

Focus on underlying factors that would cause meaningful variation in answers
across different types of people.  Avoid generic or overly broad attributes.

Respond **only** with a Python-style list of double-quoted strings.  Do not
include any explanation, headers, or prose before or after the list.

**Example output format:** ["religious affiliation", "political ideology",
"trust in government"]
```

To generate attributes from a set of questions:

### Attribute Learner: Learning Attributes from A Set of Questions

```
You are an intelligent research assistant trained to analyze survey
questions and infer which human attributes might influence how individuals
respond.

You are given a set of training-only survey questions.  Your task is to
propose a list of relevant human attributes|such as demographics, beliefs,
values, personality traits, or ideological leanings|that are likely to
affect responses to these questions.

Carefully analyze the content and framing of the questions.  Identify
underlying factors that might shape how different people respond.  Focus
on attributes that are salient, discriminative, and potentially variable
across respondents.

Respond **only** with a Python-style list of double-quoted strings.  Do not
include any explanation, headers, or prose before or after the list.

**Example output format:** ["attribute1", "attribute2", "attribute3",
"attribute4"]
```

### B.2.2. ENDOWMENT MODEL (ENDOWMENTMODEL)

System prompt:

### Endowment Model: System Prompt

```
You are an expert assistant trained to generate realistic, diverse, and
demographically plausible personas for social science surveys.

Each persona should include:

- `eid`:  a short, lowercase, variable-safe identifier that encodes key
traits (e.g., urban_liberal_30s_female).  No punctuation or spaces.

- `endow_text`:  a natural language description of the persona (1-2
sentences), written as if describing a survey respondent.

Instructions:

- Represent a wide range of age, gender, race, education, region, and
political ideology.
```

– Avoid repetition of phrasing or demographic combinations across personas.

– Do not include explanations or formatting outside of the JSON array.

To generate endowments based on attributes:

> **Endowment Model: Endowment Generation Based on Attributes**
>
> Generate {n} diverse persona(s) that vary meaningfully along the following attributes: {attr_string}.
>
> Each persona should reflect a distinct combination or value expression of these traits.
>
> Return a JSON array of dictionaries, each with:
>
> – `eid`: a short, lowercase, variable-safe identifier
>
> – `endow_text`: a brief natural-language description of the persona

To generate endowments based on survey topics (also used for Vanilla baseline):

> **Endowment Model: Endowment Generation Based on A Survey Topic**
>
> Generate {n} diverse personas for a survey experiment.
>
> Each persona must include:
>
> – eid: short, lowercase identifier
>
> – endow_text: a short natural-language description
>
> The survey topic is: {topic}
>
> Ensure diversity across demographics.
>
> Return the result as a JSON array.

### B.2.3. SURVEY CONDUCTOR (SURVEYCONDUCTOR)

To use an agent to answer a survey question:

> **Survey Conductor: Answering A Survey Question**
>
> You are completing a survey.
>
> Your answer should reflect the person described in the profile above, using their preferneces, beliefs and experiences.
>
> Respond with only the final answer string, not the code or label in brackets.
>
> Do not include any reasoning, explanation, or commentary.
>
> Do not preface your answer with phrases like 'I would choose'.
>
> Just return the answer text exactly as it appears in the options.

## B.3. Active Endowment Generation

---

**Algorithm 1** Active Endowment Generation

---

1: Initialize attribute bank $\mathcal{B}$ with core, thematic, and theoretical templates
2: Derive `survey`-specific attributes from $\mathcal{Q}_{\text{train}}$ via `AttributeLearner`; append to $\mathcal{B}$

  **Initial Sampling Stage**
3: Sample initial endowments $\mathcal{E}_0$ via equal-mode sampling from $\mathcal{B}$
4: Instantiate initial agents $\mathcal{A}_0$ from $\mathcal{E}_0$ and elicit responses $\mathcal{R}_0$ on $\mathcal{Q}_{\text{train}}$
5: Initialize `ThemeVariabilityTracker` (henceforce, `Tracker`) with $\mathcal{A}_0$, $\mathcal{R}_0$, and $\mathcal{Q}_{\text{train}}$.

  **Expansion Stage**
6: **while** agent budget $N_A$ not yet reached **do**
7:   Compute variability scores and mode sampling probabilities via `Tracker`
8:   Allocate endowment budget across modes according to probabilities
9:   **if** question patching is enabled **then**
10:    Identify lowest-entropy questions $\mathcal{Q}_{\text{low}}$ via `Tracker`
11:    **for** each $q \in \mathcal{Q}_{\text{low}}$ **do**
12:      **if** $q$ is queried for the first time **then**
13:        Extract attributes for $q$ via `AttributeLearner` and append to $\mathcal{B}$
14:      **else if** $q$ has appeared more than $n$ times **then**
15:        Enable mixed-mode strategy using $q$ and top-performing mode
16:      **end if**
17:    **end for**
18:   **end if**
19:   Generate new endowments $\mathcal{E}_{\text{new}}$ using allocated budget over sampled modes, patched question modes (if any) and mixed modes (if any).
20:   Instantiate agents $\mathcal{A}_{\text{new}}$ and elicit responses $\mathcal{R}_{\text{new}}$ on $\mathcal{Q}_{\text{train}}$
21:   Update: $\mathcal{A} \leftarrow \mathcal{A} \cup \mathcal{A}_{\text{new}}, \quad \mathcal{R}_{\mathcal{A}} \leftarrow \mathcal{R}_{\mathcal{A}} \cup \mathcal{R}_{\text{new}}$
22:   Update `Tracker` with $\mathcal{A}$, $\mathcal{R}_{\mathcal{A}}$, and $\mathcal{Q}_{\text{train}}$
23: **end while**
24: **Return:** Final agent pool $\mathcal{A}$ and full response set $\mathcal{R}_{\mathcal{A}}$

---

**Algorithm 2** Tracker Update Procedure

---

1: Input: Current agent pool $\mathcal{A}$, current agent responses $\mathcal{R}_{\mathcal{A}}$, and training questions $\mathcal{Q}_{\text{train}}$
2: **for** each question $i \in \mathcal{Q}_{\text{train}}$ **do**
3:   Compute question entropy $H_i(\mathcal{A})$
4: **end for**
5: **for** each mode **do**
6:   Compute variability score $V(\mathcal{A}_{\text{mode}})$
7: **end for**
8: Compute softmax probabilities from variability scores
9: Output: Variability scores, question entropies, and sampling probabilities.

---

## B.4. Demonstration of `P2P`: An Example Run on Wave 42

To illustrate how P2P operates in practice, we present a detailed walkthrough of a single run on ATP Wave 42 (Trust in Science, 129 questions), tracing the system from the active endowment generation through final ensemble selection.

### B.4.1. EXPERIMENTAL SETUP

Questions are split according to a $7 : 1.5 : 1.5$ ratio into the training, validation, and test sets. We set the endowment budget to be 300, with 10 endowments generated for each mode during initial sampling. Active endowment generation is run in 10 update steps with the preset attribute bank, question patching (lowest 3; 75% endowment budget), and 10 attributes drawn for each endowment generation. Gemini-2.0-flash is used as the model backend.

### B.4.2. ACTIVE ENDOWMENT GENERATION

Figure 5 displays the entropy change for each question across the update steps. Evidently, a significant number of questions experience a noticeable rise in entropy at the end of the first update step. From the second step onward, mixed mode enters the generation loop. Marginal gains in entropy are observed for the majority of questions, whereas some experience mild entropy drops. Figure 6 displays the question entropies at the end of generation.

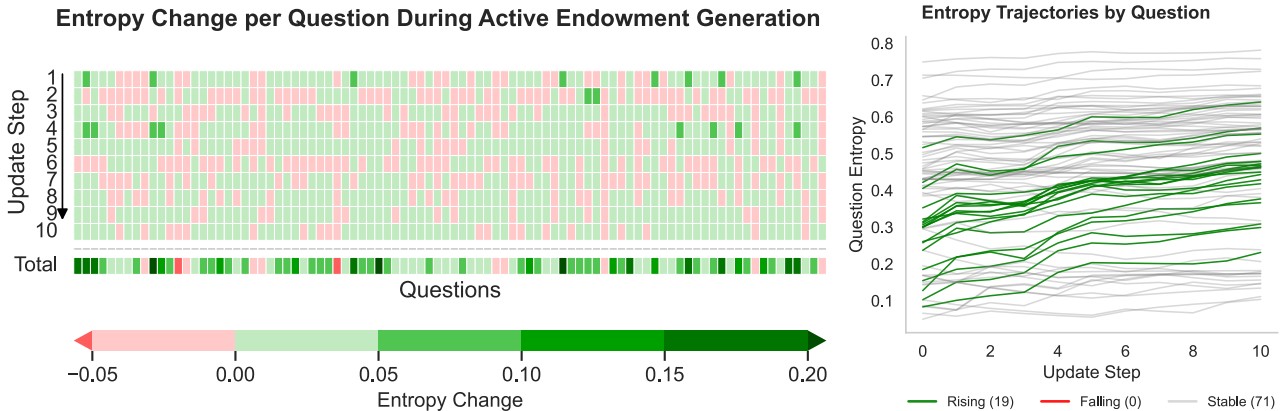

*Figure 5.* **Left:** Entropy change across update steps for each question during active endowment generation for ATP W42. Each column represents a question. Green indicates an increase in entropy from the previous step, while red indicates a decline. **Right:** Entropy trajectories by question (line chart). Active endowment generation progressively increases response diversity. Trends are classified based on slope and volatility of the entropy trajectory: "rising" if the slope exceeds 0.1 and standard deviation is above 0.02; "falling" if the slope is below –0.1 with sufficient volatility; otherwise labeled "stable".

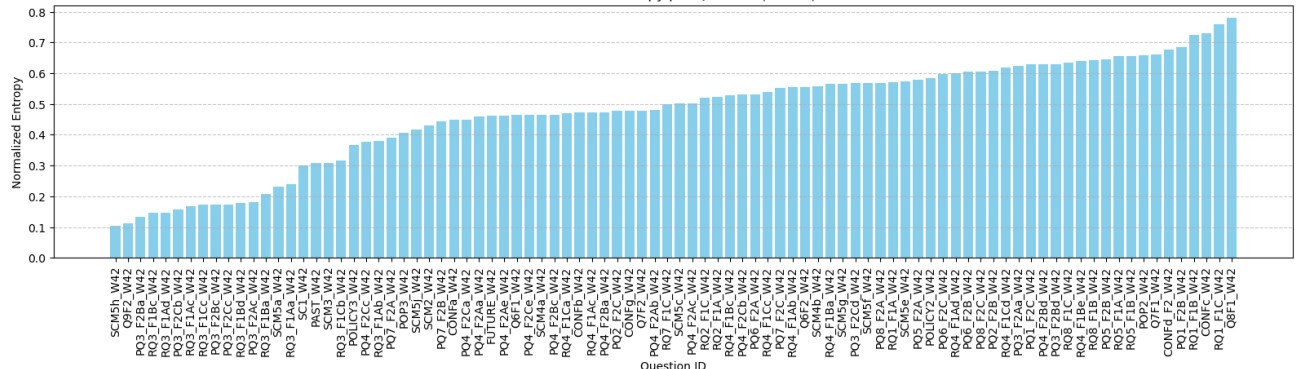

*Figure 6.* Question entropies at the end of active endowment generation.

### B.4.3. REGRESSION-BASED AGGREGATION

For regression analysis, categorical questions are binarized into question-option pairs. For the selection of the agent ensemble, we fit both a constrained lasso and a constrained elastic net, using CV to select the penalization parameters. Consequently, lasso selects 58 out of the 300 synthetic agents to form the agent ensemble, while elastic net selects 107. Figure 7 shows the hyperparameter selected for lasso using cross-validation.

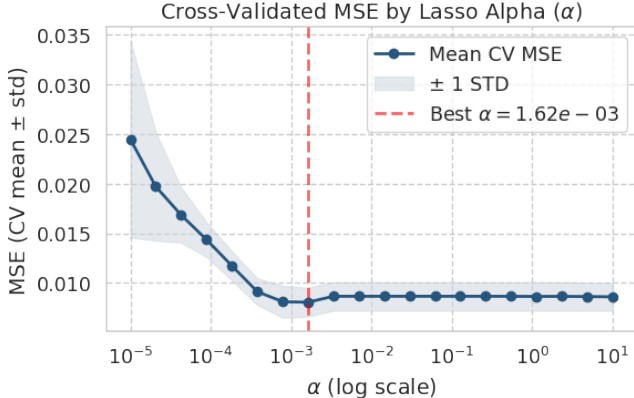

*Figure 7.* Cross-validated MSE by alpha ($\alpha$), the penalization parameter, for constrained lasso. The best alpha selected by CV is $1.62 \times 10^{-3}$.

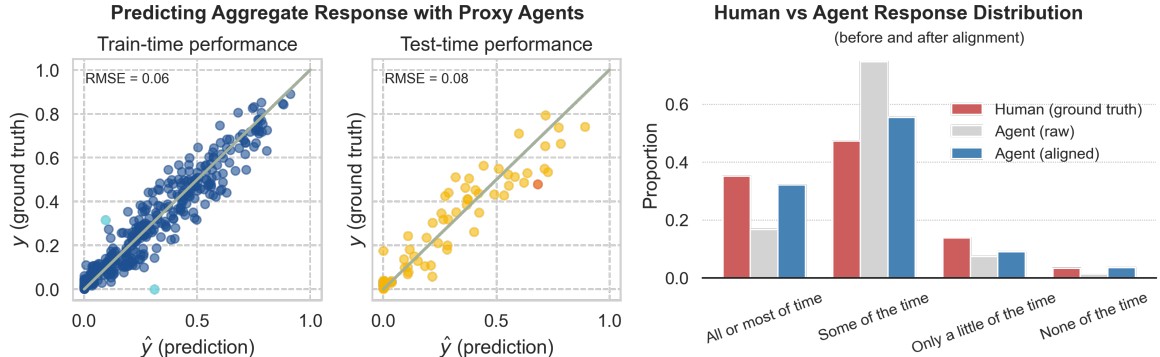

*Figure 8.* Prompts to Proxies: Emulation Results for ATP W42. **Left**: Train (left) and test (right) prediction performances for the binarized responses using constrained lasso. **Right**: Human and agent (pre- and post- alignment) aggregated responses for the sample test question `RQ4_F1Ae_W42`.

Figure 8 shows the prediction performances of the agent ensemble selected with constrained lasso. The train-time root mean squared error (RMSE) is 0.06, with test RMSE slightly higher at 0.08, indicating adequate generalization ability of the functional basis. The right panel of Figure 8 presents a snapshot for the aggregated responses to a test set question, comparing the human ground truth with agentic emulations before and after regression-based aggregation. Overall, the results are encouraging, especially considering that P2P constructs the endowments based solely on preset and learned attributes and is agnostic on the ground-truth demographic data. Figure 9 provides a breakdown of the agent ensemble selected by lasso. Notably, preset, freeform, and mixed templates all contribute non-trivially to the final ensemble. Figure 10 provides three examples of endowments generated and selected by P2P for the run on ATP W42, tracing back to their attributes, modes, and templates. Additionally, results of constrained elastic net are shown in Figures 11. Overall, the prediction results are similar to those using constrained lasso.

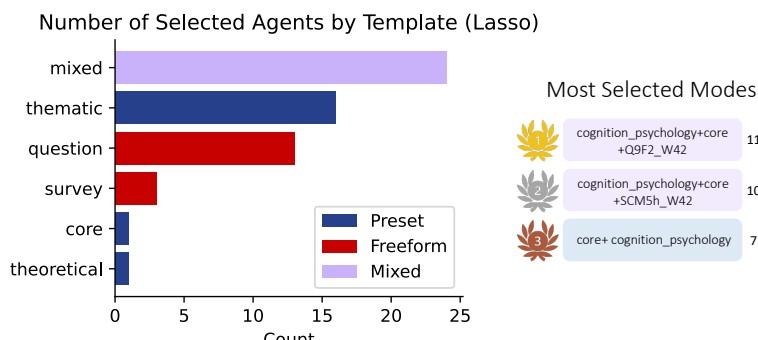

*Figure 9.* Breakdown of Selected Agents (58 in total) for W42 by Template. Preset, freeform, and mixed templates all contribute non-trivially to the final ensemble.

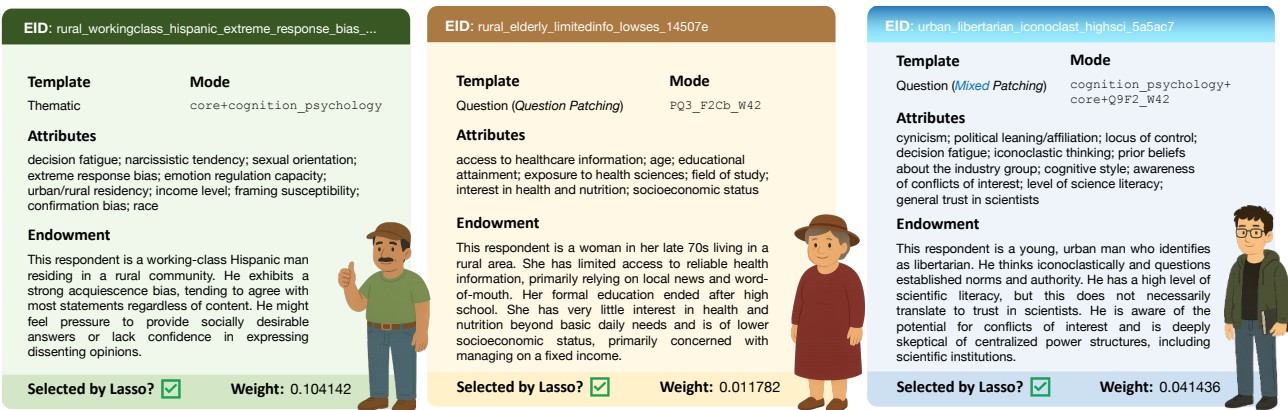

*Figure 10.* Examples of endowments generated by P2P for a run on ATP W42.

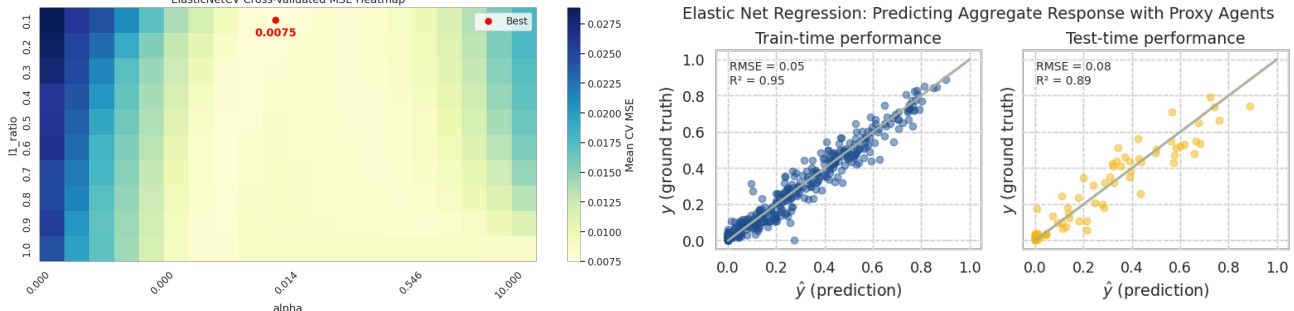

*Figure 11.* **Left:** Cross-validated MSE by alpha, the overall penalization parameter, and L1 ratio, the penalization weight for the L1 term, for constrained elastic net. The lowest cross-validated MSE is achieved at $\alpha = 6.95 \times 10^{-3}$, with l1 ratio set at $0.10$. **Right:** Prediction performance of the agent ensemble (107 selected agents) using constrained elastic net. Results are similar to those using constrained lasso.

# C. Design Validation: Simulation Studies

In order to gauge the validity of the proposed preference model and the factors affecting alignment performance, we conduct three simulation studies where we have the privilege of ground-truth knowledge.

**Simulation Design**    For the simulation studies, we use the American Trends Panel (ATP) Wave 42 (W42) as the survey source, covering the topic of public trust in science and scientists. We run the active endowment generation on the survey to get 300 persona endowments, and partition the survey questions into training, validation and test sets by the 0.7:0.15:0.15 ratio. We subsequently simulate the responses to the survey questions using endowed GPT-4o agents as the synthetic respondents. The ATP W42 dataset has in total 129 questions, which after binarization leads to 517 observations—85% of these observations form the train-and-validation (trainval) set and the rest 15% is used to test the predictive accuracy of the selected agent ensemble.

**Construction of the Ground Truth Data**    To construct the ground truth data, we sample a fraction of the endowments as the ground truth and randomly assign weights to these endowments, which represent the proportions of the associated personas in the ground truth population. The ground truth data—percentages of individuals choosing difference options for a given question—are computed using weighted averages of responses provided by the ground-truth agents. The unselected endowments are used to construct the pool of proxy agents.

**Regression Method**    For the simulation studies, we choose constrained lasso as our main regression alignment method for investigation. As a secondary choice, constrained elastic net is an extension of constrained lasso that trades off between the L1 and L2 penalty terms. It is expected to have more stable performance when the features (agent responses) are highly correlated. Our initial simulations indicate that constrained elastic net demonstrates near identical performance to constrained Lasso, partly because the latter is a special case of the former and we are using cross-validation to select the best penalization parameters. Therefore, for the sake of building intuitions, we focus on constrained lasso in the simulations and expect the results to be largely indicative of constrained elastic net's performance.

## C.1. Recovering Ground Truth Agents with Constrained Lasso

As a first test of constrained lasso as a variable selection algorithm, we aim to evaluate its ability to recover ground-truth agents from a broader candidate pool. To render the selection task nontrivial, we set the prevalence of the ground-truth agents to 0.3—out of 10 selectable agents, only 3 contribute to the construction of the ground-truth signal. As the proxy agents dominate the agent pool, an adequate number of observations becomes essential for Lasso to reliably identify the true contributors. As a key metric, we define **observation-to-agent ratio (OAR)** to be the ratio of the number of training and validation observations to the number of selectable agents. By default, the OAR for our simulation baseline is 1.203.

To develop a panoramic view of constrained lasso's ability in ground truth recovery, we conduct multiple simulation rounds under varying OARs. We generate different OAR variations using two complementary subsampling strategies:

- **Subsampling endowments**. To increase the OAR, we adopt a subsampling strategy on endowments, where we subsample from the pool of selectable endowments, varying the total number of candidate agents while keeping the proportion of ground-truth agents fixed at 30%. The ground-truth weights are renormalized to sum to 1. We vary the subsample fraction from 0.5 to 1 in 20 equally spaced steps.

- **Subsampling questions**. To decrease the OAR, we subsample survey questions while keeping the original train-validation-test split and ratio unchanged. This mimics a setting with reduced behavioral signal for inference. As in the endowment-based approach, we use 20 subsample fractions ranging from 0.5 to 1.

We repeat each subsampling procedure 10 times to construct the final dataset used in our simulation study. Figure 12 shows the trends of the mean squared error of the test set and the precision—that is, the proportion of selected agents who possess ground-truth endowments—across different OARs. To further illustrate the dynamics of selection, Figure 13 presents three representative snapshots of simulation runs, corresponding to low, medium, and high OAR settings.

As the figures indicate, constrained lasso's ability to recover ground-truth agents from the candidate pool improves with a higher observation-to-agent ratio. This is intuitive as more observations offer a richer signal of both individual-level idiosyncrasies and aggregate preferences. With abundant behavioral cues, the selection algorithm is better equipped to

distinguish true contributors from correlated siblings. In contrast, when observations are sparse, Lasso tends to relinquish its sharpness as a selection tool and instead defaults to identifying a functional basis that best explains the limited data. The mild decline in predictive accuracy under low OAR regimes suggests that this functional basis still approximates the aggregate preference reasonably well, albeit imperfectly.

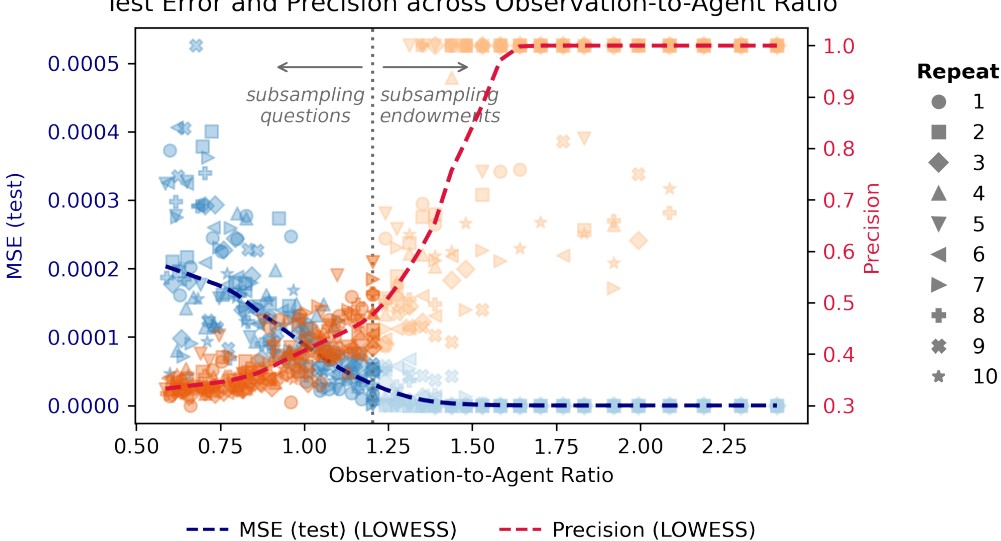

*Figure 12.* Mean Squared Error (Test Set) and Precision by Different Observation-to-Agent Ratio (OAR) for Constrained Lasso. Each simulation is repeated 10 times with different random seeds. Subsampling of endowments is done with fixed ground-truth-to-proxy ratio. Subsampling of responses is done with fixed train-valid-test split and ratio.

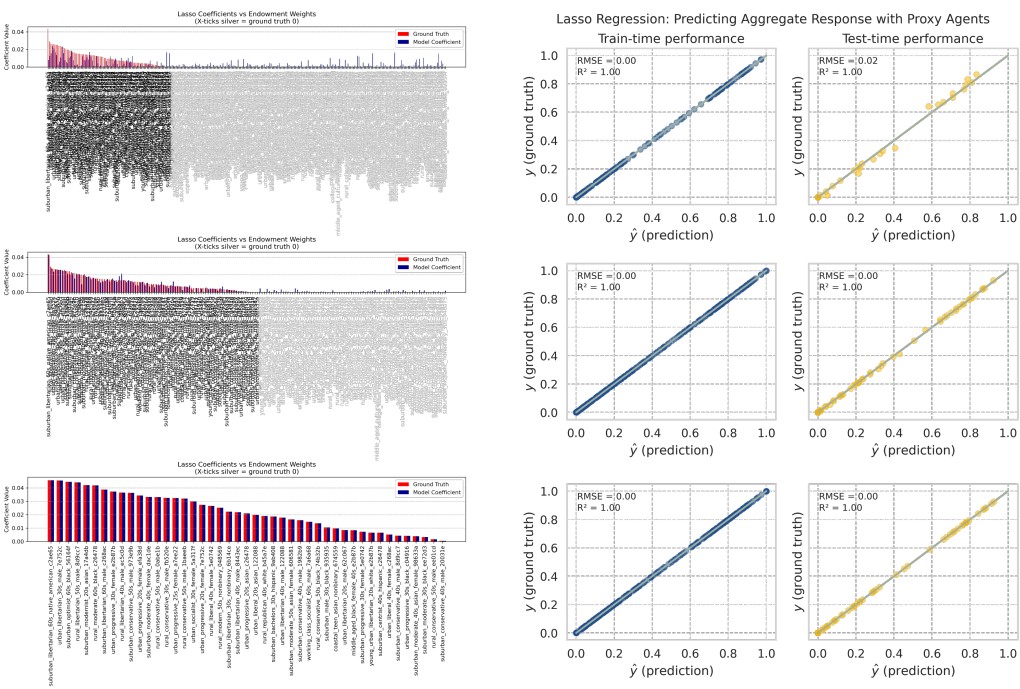

*Figure 13.* Snapshots of Weight-vs-Coefficient Comparison and Prediction Performance. Each row represents a simulation run. The red bars on the left panel represent ground truth agent weights used to generate the aggregate data and the blue bars Lasso coefficients. The observation-to-agent ratios for the top, middle and bottom panels are: 0.617, 1.203, 2.407. Lasso's ability to recover the ground truth agents from the agent pool degrades with lower observation-to-agent ratio, but its predictive accuracy only drops mildly.

### C.2. Emulating Ground Truth Responses with Proxy Agents

While the first simulation exercise has readily allured to constrained lasso's ability to find a functional basis to explain the observed data, in this second simulation study we extend the investigation by removing the ground-truth agents from the selectable agent pool. Under this setting, constrained lasso needs to fully rely on proxy agents to recover the ground-truth patterns. We analyze two distinct scenarios:

- **Subsampling proxies.** We fix the total number of observations and reduce the proxy pool by retaining only a fraction as selectable agents. We sweep across the fractions from 0.05 to 1 in 20 equal spaced steps.

- **Subsampling training and validation questions.** We fix the proxy pool but retain a fraction of training and validation observations. Similar to subsampling proxies, we construct 20 simulation rounds gradually raising the retained fraction from 0.05 to 1.

Like in Simulation Study 1, we repeat each simulation 10 times using different random seeds to form the final results. In Figure 14, the left panel displays the mean squared errors (MSE) and coefficients of determination ($R^2$) for the subsampling proxies scenario, while the right panel showcases the associated metrics for the subsampling training and validation questions scenario.

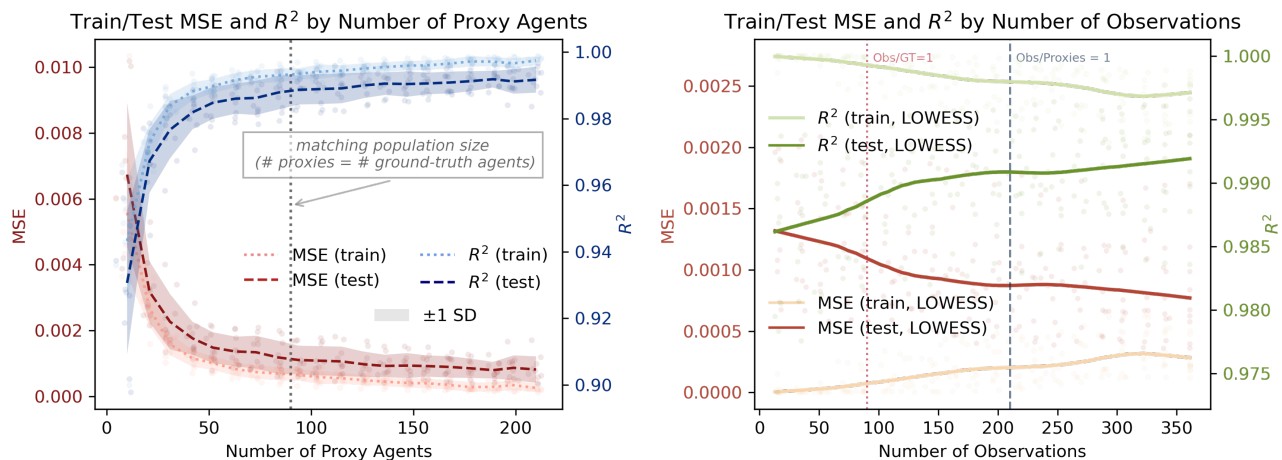

*Figure 14.* Left: Mean Squared Errors and Coefficients of Determination ($R^2$) by Number of Proxy Agents. Right: Mean Squared Errors and Coefficients of Determination ($R^2$) by Number of Observation. Each simulation is repeated 10 times with different random seeds.

Notably, constrained lasso exhibits robust performance across all simulated conditions. Even with only 10 selectable proxy agents, it achieves a mean test MSE of 0.0067, yielding an $R^2$ of 0.93. Predictive accuracy improves as we increase the number of selectable proxy agents, plateauing at 0.99. In comparison, decreasing the number of observations has a lesser effect on predictive performance: in the recorded run with the lowest number of observations at 13, constrained lasso attains a test MSE of 0.0019 and an $R^2$ of 0.98. Figure 15 displays representative snapshots of the simulation runs.

The high predictive accuracy of the simulation runs indicates that constrained lasso can efficiently find a functional basis from a select few agents and determine the agent weights to emulate the ground-truth data patterns based on limited observations. This is in fact not surprising because in the initial assignment of the endowments into the ground truth and proxy groups we have used a random partition. As informed by our proposed behavioral preference model, this causes the ground truth vectors—used in the simulation study to form the aggregate preference—and proxy vectors to span the same preference space. As the preference space is not directly accessible, we can derive a partial gauge through the observed agent responses. If the responses of a group of agents have higher variability, it is an indication that the preference space they span is large. Conversely, if agents tend to agree on their responses to different questions, it is a signal that their associated vectors in the preference space might overlap in sub-dimensions, leading to a pool generalization ability.

To effectively measure response variability, in the paper we introduce the notion of **question entropy**:

$$H_i(\mathcal{A}) = \frac{-\sum_{k=1}^{K_i} p_{ik} \log_2 p_{ik}}{\log_2 K_i} \tag{11}$$

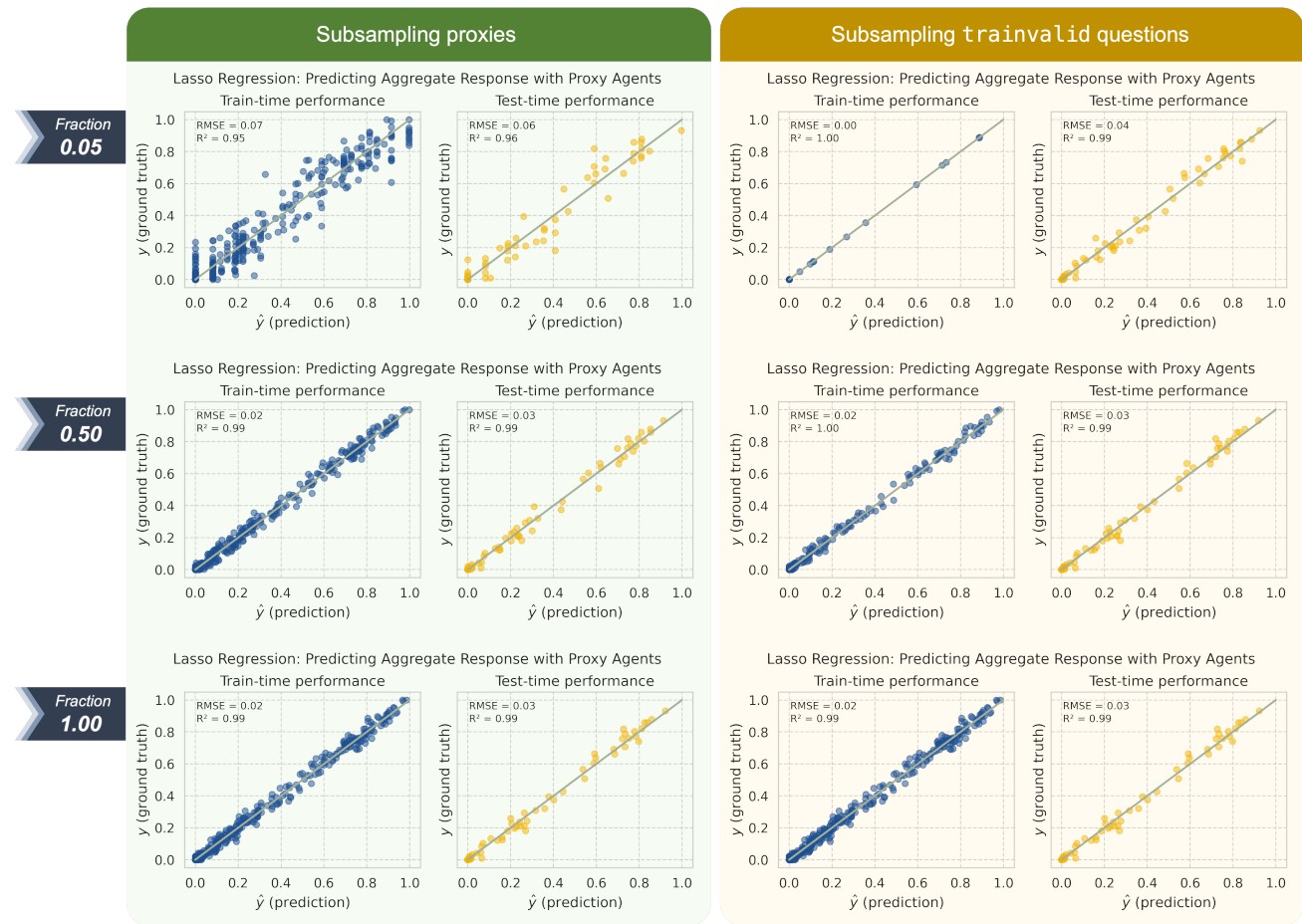

*Figure 15.* Snapshots of Prediction Performance for Simulation Study 2.

where $H_i(\mathcal{A})$ is the normalized entropy of question $i$ based on responses from a group of agents $\mathcal{A}$, $K_i$ denotes the number of unique response options $\{1, \ldots, K_i\}$ for question $i$, and $p_{ik}$ represents the empirical proportion of responses selecting option $k$. The normalization factor ensures comparison across questions. A high question entropy indicates that the responses are highly varied for this question.

For the simulation studies, we define **group entropy** as the average question entropy computed using responses from that group:

$$H(\mathcal{A}) = \frac{1}{N} \sum_{i=1}^{N} H_i(\mathcal{A}). \tag{12}$$

The group entropy is calculated using responses to all survey questions, regardless of the split. Based on it, we introduce the notion of **Entropy Coverage Ratio (ECR)** defined as the ratio between the group entropy of proxy agents and that of the ground-truth agents:

$$ECR = \frac{H(\mathcal{A}_{\text{proxy}})}{H(\mathcal{A}_{\text{gt}})}. \tag{13}$$

An ECR lower than 1 indicates that the proxy agents have less response variability than the ground truth, while an ECR greater than 1 indicates the opposite.

The left panel of Figure 16 shows the distribution of entropy coverage ratios (ECRs) for the two analyzed scenarios. The mass is concentrated around 1.0, corroborating the earlier observation that random partitioning tends to preserve entropy structure. The right panel plots ECR against the number of proxies for the proxy-subsampling scenario. The relationship between ECR and the number of proxies is subtly positive: when the number of proxies is small, the likelihood of drawing

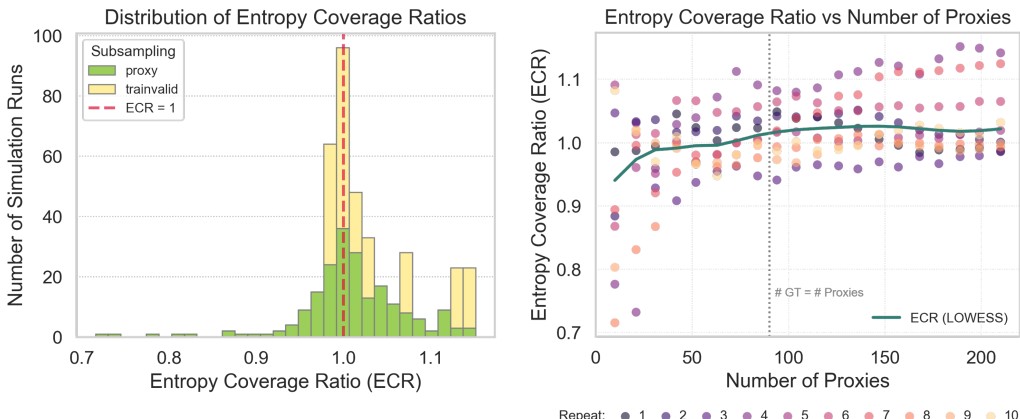

*Figure 16.* Entropy Coverage Ratio Diagnostics for Simulation Study 2.

an unrepresentative sample increases. As the sample size grows, group-level entropy converges toward the population-level entropy—an empirical manifestation of the law of large numbers.

### C.3. Response Variability Matters

In this final simulation study, we analyze the effect of the entropy coverage ratio on predictive accuracy for constrained lasso.

Figure 17 displays the group entropies for THE W42 endowments organized by modes—we call them **mode entropies** in the endowment generation logic. We subsume them further into three different tiers according to the group entropy value: in total, we have 71 endowments in the low entropy tier, 92 in the mid entropy tier, and 137 in the high entropy tier.

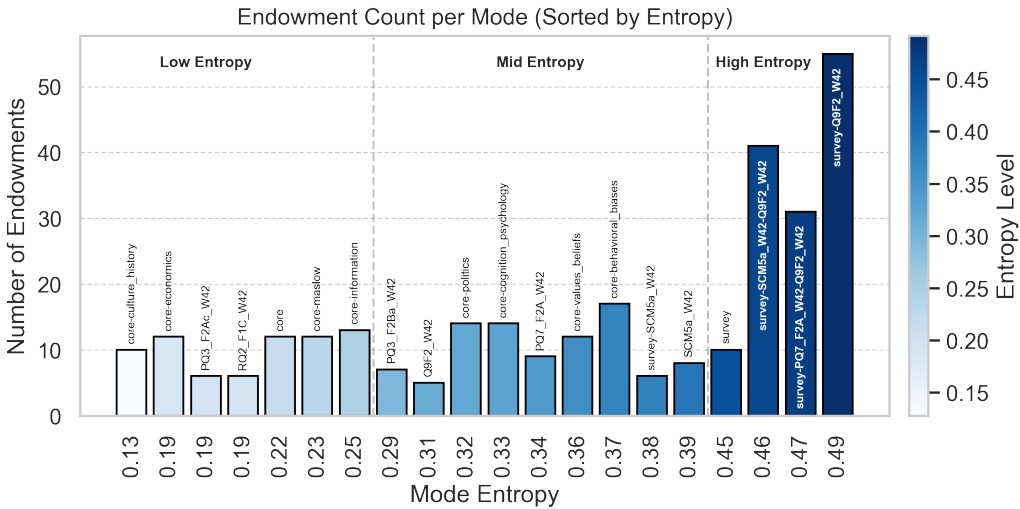

*Figure 17.* Endowment Count per Mode (Sorted by Entropy).

To cover a wide range of ECR ratios, we design the following simulation strategy:

1. **Ground Truth Construction.** For each simulation round, pick one of the three tiers as the ground-truth tier and sample 30 endowments from the tier to construct the ground-truth agents and assign ground-truth weights. The unselected endowments from the tier are joined with the endowments from the other two tiers to form the pool of proxy candidates.

2. **Proxy Agents Construction.** For each simulation run, we select 30 proxy agents to form the proxy pool. We begin by selecting agents from the lowest entropy modes (ensuring that the cumulative candidate endowments surpass 30), and

in each consecutive run, include the next mode with higher entropy into the selectable modes. We sweep across the modes by increasing mode entropy.

3. **Proxy Agents Selection and Response Emulation.** In each simulation run, after the proxy pool has been constructed, we use constrained lasso to select proxy agents to emulate the observed aggregate response data.

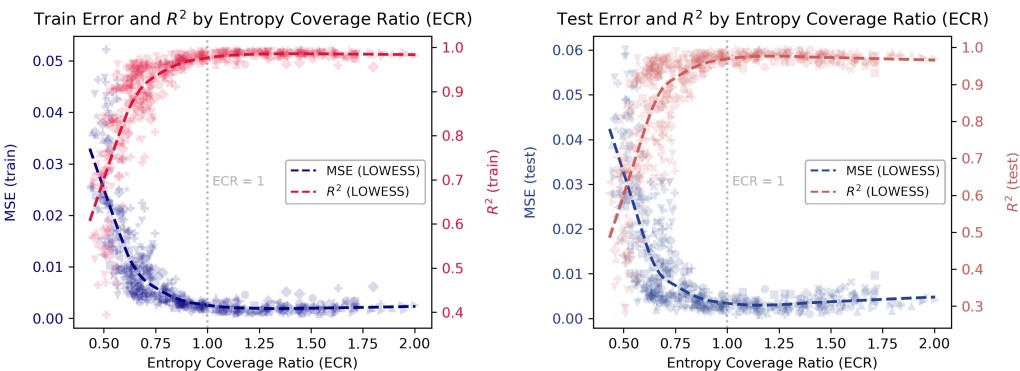

*Figure 18.* Mean Squared Error and Coefficient of Determination ($R^2$) by Entropy Coverage Ratio (ECR) for the trainval set (left) and test test (right). Entropy Coverage Ratio is defined as the ratio of the average response entropies of the proxies and ground-truth agents. LOWESS curves are fit using 10 repetitions.

We repeat each simulation 10 times to form the final results. Figure 18 visualizes the train and test performance of the constrained lasso by the entropy coverage ratio. As the ECR increases, the predictive accuracy improves at both the training and the test times, lending further credence to the analysis based on the preference model in the second simulation study.

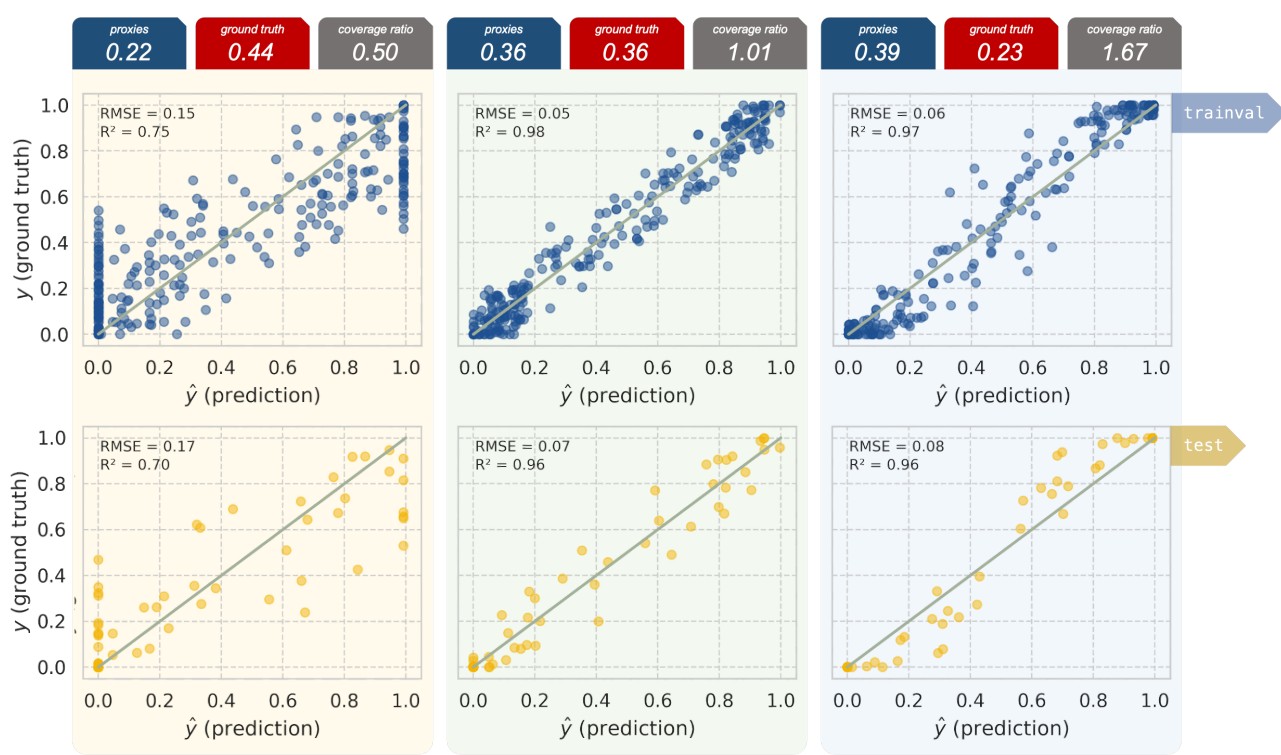

*Figure 19.* Snapshots of Prediction Performance for Simulation Study 3.

Figure 19 offers three snapshots of the simulation runs corresponding to ECRs equaling 0.50, 1.01 and 1.67. When proxy entropy is low compared with ground truth, the proxy agents fail to provide a sufficiently expressive basis to span the

aggregate ground-truth preference. As a result, constrained Lasso can only approximate a lower-dimensional projection of the true preference signal. This results in poor prediction performance, as illustrated in the leftmost snapshot. Conversely, when proxy entropy exceeds ground-truth entropy, prediction error begins to increase—albeit at an infinitesimal scale. The subtle deterioration of performance when ECR surpasses 1 could be due to various reasons. A plausible cause is from the data perspective: when the proxy pool becomes too diverse, it may introduce spurious variability that does not align well with the true underlying structure. Put differently, an observed response might amalgamate signals from different latent preference factors, some of which are irrelevant to the aggregate preference in focus. While greater diversity expands the representational capacity of the proxy pool, it may also dilute the signal and induce overfitting, thereby reducing generalization accuracy. In practice, we posit that over-diversity is less likely than under-diversity, as existing studies using demographic conditioning (Argyle et al., 2023; Park et al., 2024) indicate LLM agents do not match closely with human ground truth. This justifies optimizing response variability in Stage 1 of P2P.

The key takeaway from the simulation study is that absent the knowledge of ground truth, it is crucial to ensure an adequate group entropy among the proxy agents to form a functional basis for ground-truth emulation. However, one should also refrain from solely relying on entropy as a measure of preference diversity lest spurious correlations beguile us under limited data regimes. While our alignment method is intentionally designed to function with minimal data—requiring only aggregate response data—in practice, if individual human responses are available, a practitioner may use the group entropy computed from these responses as an anchor for endowment generation.

# D. Empirical Experiments: Supplementary Results

### D.1. Main Results: Baseline Comparison on ATP Waves

In our setup, MSE is computed on the option-level probabilities after binarizing each multiple-choice item, so MSE penalizes misallocation of probability mass across the full categorical distribution. To facilitate comparisons across works and corroborate the use of MSE as a valid distribution metric, we follow previous studies (Cao et al., 2025; Feng et al., 2024; Durmus et al., 2024) and additionally report Jensen-Shannon Divergence (1-JSD) and Earth Mover Distance (EMD) for the key results in this paper.

Table 3 presents the full distribution metrics comparison of baselines and P2P across 14 ATP waves. P2P consistently outperforms Vanilla and PERSONA across all three metrics, registering lower MSE (0.014 vs. 0.026–0.029), higher 1-JSD (0.828 vs. 0.757–0.763), and lower EMD (0.067 vs. 0.144–0.178). The 53% reduction in EMD is particularly notable, indicating that P2P predictions are structurally closer to ground-truth distributions.

*Table 3.* Full distribution metrics comparison across 14 ATP waves. MSE measures mean squared error of predicted response probabilities. 1-JSD (1 minus Jensen-Shannon Divergence) measures distributional similarity (higher is better). EMD (Earth Mover's Distance) measures distributional distance (lower is better). P2P reports mean $\pm$ std over 3 runs.

| | Vanilla | | | PERSONA | | | P2P | | |
|---|---|---|---|---|---|---|---|---|---|
| Wave | MSE | 1-JSD | EMD | MSE | 1-JSD | EMD | MSE | 1-JSD | EMD |
| W26 | .015 | .817 | .075 | .021 | .782 | .078 | $.010_{\pm.000}$ | $.868_{\pm.001}$ | $.056_{\pm.001}$ |
| W27 | .030 | .766 | .419 | .042 | .730 | .556 | $.022_{\pm.001}$ | $.823_{\pm.006}$ | $.072_{\pm.003}$ |
| W29 | .030 | .757 | .096 | .030 | .743 | .107 | $.016_{\pm.005}$ | $.815_{\pm.027}$ | $.083_{\pm.013}$ |
| W32 | .020 | .803 | .310 | .026 | .768 | .454 | $.016_{\pm.002}$ | $.833_{\pm.007}$ | $.066_{\pm.003}$ |
| W34 | .022 | .763 | .091 | .020 | .779 | .086 | $.011_{\pm.001}$ | $.822_{\pm.010}$ | $.070_{\pm.007}$ |
| W36 | .033 | .750 | .087 | .035 | .761 | .081 | $.019_{\pm.004}$ | $.818_{\pm.013}$ | $.063_{\pm.006}$ |
| W41 | .022 | .759 | .098 | .023 | .777 | .092 | $.013_{\pm.001}$ | $.828_{\pm.002}$ | $.069_{\pm.006}$ |
| W42 | .034 | .758 | .086 | .038 | .750 | .091 | $.011_{\pm.000}$ | $.853_{\pm.003}$ | $.053_{\pm.001}$ |
| W45 | .016 | .804 | .275 | .028 | .778 | .482 | $.008_{\pm.002}$ | $.848_{\pm.013}$ | $.057_{\pm.010}$ |
| W49 | .045 | .731 | .095 | .047 | .731 | .090 | $.021_{\pm.004}$ | $.814_{\pm.017}$ | $.065_{\pm.005}$ |
| W50 | .027 | .708 | .103 | .024 | .711 | .111 | $.013_{\pm.004}$ | $.805_{\pm.015}$ | $.076_{\pm.008}$ |
| W54 | .022 | .734 | .092 | .026 | .740 | .083 | $.014_{\pm.002}$ | $.801_{\pm.019}$ | $.068_{\pm.007}$ |
| W82 | .025 | .742 | .096 | .019 | .778 | .079 | $.008_{\pm.002}$ | $.856_{\pm.012}$ | $.052_{\pm.005}$ |
| W92 | .025 | .784 | .093 | .031 | .775 | .096 | $.017_{\pm.003}$ | $.810_{\pm.008}$ | $.083_{\pm.004}$ |
| **Avg** | .026 | .763 | .144 | .029 | .757 | .178 | **.014** | **.828** | **.067** |

Table 4 shows stability analysis across four representative waves spanning weak, medium and strong alignment difficulty (characterized by test MSE). P2P ranks top on entropy, MSE, 1-JSD and EMD across all waves, with cost consistently between Vanilla and PERSONA. Baseline variance is negligible for MSE (max std = 0.005), entropy (max std = 0.007), 1-JSD (max std = 0.012), and cost (max std = 0.007). EMD shows higher baseline variance on W45 (std up to 0.227 for PERSONA, 0.125 for Vanilla), indicating occasional distributional instability that P2P avoids (max std = 0.010). The performance gap substantially exceeds error bands in most cases, confirming that improvements are statistically meaningful. We report 3 runs for P2P to verify robustness of the active endowment generation process, and single runs for baselines given their low variance.

Table 5 compares lasso and elastic net as Stage 2 regression choices for P2P. Performance is nearly identical across all waves (average $\Delta$ MSE $\approx$ 0.0001, with 10 of 14 waves showing no difference). As lasso typically selects a more compact ensemble than elastic net, we prioritize its use as the default regression method.

#### D.1.1. CLUSTERING

Based on Vanilla and PERSONA, we design two variants using K-means clustering as a pre-processing step. Specifically, we first expand the initial pool of personas for Vanilla and PERSONA to 1000, computing their embeddings (via OpenAI's text-embedding-3-large) and cluster them (k=300). We pick the personas (medoids) closest to the centroids to construct the candidate pool and then run our regression stage on the resulting pool. Table 6 shows results on ATP W42.

Clustering yields a test MSE of 0.0266 for Vanilla that lies well within the variability band of the original Vanilla baseline (0.0285 $\pm$ 0.0049) and slightly worse MSE for PERSONA (0.0395 vs. 0.0362 $\pm$ 0.0011), with no conspicuous change

*Table 4.* Stability analysis on representative ATP waves. We select waves representing weak (W27), medium (W32), and strong (W42, W45) alignment performance. All values are mean ± std over 3 runs. Standard deviations are consistently small across all metrics, justifying single-run reporting for baselines in Table 1.

| Wave | Method | Entropy ↑ | MSE ↓ | 1-JSD ↑ | EMD ↓ | Cost ($) ↓ |
|------|--------|-----------|-------|---------|-------|------------|
| W27 (Weak) | Vanilla | .462 ± .007 | .032 ± .005 | .795 ± .008 | .409 ± .019 | **0.63** ± .004 |
| | PERSONA | .402 ± .000 | .041 ± .001 | .728 ± .004 | .557 ± .011 | 1.26 ± .000 |
| | P2P | **.536** ± .010 | **.022** ± .001 | **.823** ± .006 | **.071** ± .003 | 0.71 ± .004 |
| W32 (Medium) | Vanilla | .551 ± .006 | .019 ± .001 | .801 ± .003 | .350 ± .001 | **0.64** ± .004 |
| | PERSONA | .557 ± .000 | .025 ± .000 | .772 ± .004 | .444 ± .008 | 1.28 ± .000 |
| | P2P | **.618** ± .015 | **.016** ± .002 | **.833** ± .007 | **.066** ± .003 | 0.72 ± .001 |
| W42 (Strong) | Vanilla | .317 ± .007 | .029 ± .005 | .768 ± .010 | .081 ± .004 | **0.86** ± .007 |
| | PERSONA | .287 ± .000 | .036 ± .001 | .753 ± .003 | .090 ± .001 | 1.71 ± .000 |
| | P2P | **.445** ± .026 | **.010** ± .002 | **.863** ± .019 | **.048** ± .007 | 0.97 ± .004 |
| W45 (Strong) | Vanilla | .389 ± .002 | .015 ± .003 | .805 ± .012 | .146 ± .125 | **0.62** ± .004 |
| | PERSONA | .365 ± .001 | .029 ± .001 | .773 ± .006 | .219 ± .227 | 1.24 ± .000 |
| | P2P | **.445** ± .006 | **.008** ± .002 | **.848** ± .013 | **.057** ± .010 | 0.69 ± .002 |

*Table 5.* Comparison of lasso and elastic net regression for P2P across 14 ATP waves. Values are mean ± std over 3 runs. Both methods yield nearly identical performance, demonstrating robustness to regularization choice.

| Wave | Lasso MSE | ENet MSE | Δ | Winner |
|------|-----------|----------|---|--------|
| W26 | .0098 ± .0002 | .0098 ± .0002 | .0000 | Tie |
| W27 | .0218 ± .0010 | .0238 ± .0027 | .0020 | Lasso |
| W29 | .0156 ± .0049 | .0156 ± .0049 | .0000 | Tie |
| W32 | .0160 ± .0019 | .0160 ± .0019 | .0000 | Tie |
| W34 | .0111 ± .0011 | .0111 ± .0011 | .0000 | Tie |
| W36 | .0191 ± .0042 | .0187 ± .0037 | .0004 | ENet |
| W41 | .0126 ± .0009 | .0126 ± .0009 | .0000 | Tie |
| W42 | .0110 ± .0003 | .0101 ± .0006 | .0009 | ENet |
| W45 | .0079 ± .0023 | .0079 ± .0023 | .0000 | Tie |
| W49 | .0211 ± .0037 | .0211 ± .0037 | .0000 | Tie |
| W50 | .0133 ± .0039 | .0133 ± .0039 | .0000 | Tie |
| W54 | .0144 ± .0018 | .0146 ± .0018 | .0002 | Lasso |
| W82 | .0083 ± .0016 | .0083 ± .0016 | .0000 | Tie |
| W92 | .0168 ± .0029 | .0173 ± .0023 | .0005 | Lasso |
| **Avg** | **.0142** | **.0143** | .0001 | – |

*Table 6.* Comparison of Vanilla, PERSONA and their clustering-based variants where 1000 agents are reduced to 300 via $k$-means clustering.

| **Metric** | **Vanilla** | **Vanilla (1000→300)** | **PERSONA** | **PERSONA (1000→300)** |
|------------|-------------|------------------------|-------------|------------------------|
| Avg. Entropy ↑ | 0.3172 ± 0.0069 | 0.3162 | 0.2871 ± 0.00026 | 0.2844 |
| Test MSE (Lasso) ↓ | 0.0285 ± 0.00490 | 0.0266 | 0.0362 ± 0.00116 | 0.0395 |
| Test MSE (Enet) ↓ | 0.0286 ± 0.00476 | 0.0266 | 0.0339 ± 0.00124 | 0.0387 |
| Cost (USD) ↓ | **0.8649** ± **0.0070** | 0.8834[†] | 1.7129 ± 0.000016[*] | 1.7203[*†] |

[*] Only considering survey elicitation cost on 300 personas—actual cost is higher.
[†] Excluding embedding cost.

on entropy and a slight increase in cost (for Vanilla due to extra generation). Hence, clustering does not yield a stronger baseline.

This matches our intuition for two reasons. First, when we apply clustering on top of the existing baselines (starting from the same 300 endowments/personas), it can only discard information that the regression step could have used. In other words, it weakens the candidate pool without giving L1 regression any new capability. Second, even in the more lenient setting where we give clustering a head start by generating 1000 candidates and then compressing them to 300, the mechanism still does not produce a stronger baseline. As clustering operates in the embedding space and tends to group personas by textual or stylistic similarity, it does not necessarily track preference-relevant variation in survey responses. As a result, minority or low-density preference types are prone to being averaged away in embedding clustering, whereas our regression layer selects proxies directly based on their contribution to matching the observed response distributions. This corroborates the design of our main baselines.

D.1.2. RAW MODEL PERFORMANCE

To better contextualize the gain of the ensemble methods (Vanilla, PERSONA, and `P2P`), we provide the raw model backend performance as a reference baseline. To construct the baseline, we directly query the model backend (gemini-2.0-flash) for answers to the survey questions, using the following instruction:

```
You are completing a public opinion survey.  Answer each question as an
average adult respondent in the US population, based only on the question
text and options.  Respond with only the final answer string, not the
code or label in brackets.  Do not include any reasoning, explanation, or
commentary.  Do not preface your answer with phrases like 'I would choose'.
Just return the answer text exactly as it appears in the options.
```

We draw 300 Monte-Carlo samples per question with temperature set to 0.7. This results in an empirical answer distribution per question. On ATP W42, this baseline attains test MSE 0.0918 and average entropy 0.0161, conspicuously worse than the results for the ensemble baselines, confirming the gains via ensemble methods (Vanilla, PERSONA, and `P2P`).

**D.2. Locale Generalization: World Values Survey**

The main paper results focus on the American Trends Panel (ATP), which represents a single survey program targeting the US population. While our 14-wave study demonstrates that `P2P` generalizes across multiple topics and different cross sections, it remains intriguing how it performs at other locales and on different survey programs. To address this question, we evaluate `P2P` on the World Values Survey (WVS) Wave 7 (Haerpfer et al., 2022), using cleaned data from Cao et al. (2025). We select three locales representing different cultural and linguistic contexts: **United States**, as it provides a bridge to our ATP findings, enabling direct comparison within the same population under a different survey program; **Great Britain**, which represents Western Europe, sharing linguistic and value overlap with the US but differing in political and cultural contexts; and **Hong Kong SAR**, a multicultural hub in East Asia, allowing us to test on populations less represented in typical LLM training data.

D.2.1. EXPERIMENT SETUP

For the WVS experiment, we use a similar setup to the ATP study: survey questions are split randomly according to a $7 : 1.5 : 1.5$ ratio into the training, validation, and test sets; endowment budget set is set at 300; initial sampling generates 10 endowments for each mode; active endowment generation is run in 5 update steps with the preset attribute bank, question patching (lowest 3; 75% endowment budget); 10 attributes drawn for each endowment generation; Gemini-2.0-flash is used as the backend. Different from the ATP study, we additionally pass the locale information (US, GB, HK) to the endowment model to ensure regional relevance of generated personas.

Notably, our random split protocol differs from Cao et al. (2025)'s Q1+Q2→Q3 topic shift protocol (Appendix D.4). This is intentional, as it isolates the effect of locale from confounding topic shift effects, providing a cleaner assessment of cross-locale generalizability.

Table 7 reports results across the three locales. Each experiment is repeated 3 times.

### D.2.2. RESULTS AND DISCUSSION

*Table 7.* P2P performance on WVS Wave 7 (US, Great Britain, Hong Kong SAR) under a 7:1.5:1.5 question split, using gemini-2.0-flash as the backend. Reported are mean ± std over 3 repeated runs.

| Metric | WVS W7 US | WVS W7 GB | WVS W7 HK |
|---|---|---|---|
| Avg. Entropy ↑ | 0.5720 ± 0.0149 | 0.5710 ± 0.0044 | 0.5478 ± 0.0211 |
| Cost (USD) ↓ | 1.7551 ± 0.0097 | 1.7095 ± 0.0031 | 1.7365 ± 0.0080 |
| Test MSE (Lasso) ↓ | 0.0149 ± 0.0005 | 0.0141 ± 0.0014 | 0.0242 ± 0.0013 |
| Test 1-JSD (Lasso) ↑ | 0.8030 ± 0.0055 | 0.7895 ± 0.0049 | 0.7590 ± 0.0046 |
| Test MCPA (Lasso) ↑ | 0.6667 ± 0.0152 | 0.6481 ± 0.0321 | 0.5789 ± 0.0263 |
| Test EMD (Lasso) ↓ | 0.1075 ± 0.0035 | 0.1038 ± 0.0081 | 0.1424 ± 0.0034 |
| Test MSE (Elastic Net) ↓ | 0.0147 ± 0.0005 | 0.0137 ± 0.0010 | 0.0244 ± 0.0005 |
| Test 1-JSD (Elastic Net) ↑ | 0.8066 ± 0.0067 | 0.7950 ± 0.0073 | 0.7602 ± 0.0032 |
| Test MCPA (Elastic Net) ↑ | 0.6930 ± 0.0152 | 0.6296 ± 0.0321 | 0.5877 ± 0.0304 |
| Test EMD (Elastic Net) ↓ | 0.1066 ± 0.0025 | 0.1004 ± 0.0054 | 0.1430 ± 0.0046 |

*Table 8.* Comparison of P2P performance on ATP (US) and WVS (US). ATP values are mean ± std over 14 waves; WVS values are mean ± std over 3 runs.

| Metric | ATP (14 waves) | WVS US |
|---|---|---|
| Test MSE (Lasso) ↓ | 0.014 ± 0.004 | 0.015 ± 0.001 |
| Avg. Entropy ↑ | 0.53 ± 0.06 | 0.57 ± 0.01 |
| Cost (USD) ↓ | 0.8 ± 0.2 | 1.76 ± 0.01 |

As shown in Table 7, performance is higher for US (MSE ≈ 0.015) and GB (MSE ≈ 0.014) and somewhat lower for HK (MSE ≈ 0.024), aligning with broader findings that large language models trained primarily on English and Western-centric data tend to better match Western public opinion than non-Western populations (Durmus et al., 2024). The HK performance gap motivates the error analysis presented in Section 4.4 of the main paper. Despite the gap, the HK results remain within the performance range observed across our 14-wave ATP study presented in the main paper (cf. Table 1), suggesting that while locale affects difficulty, P2P remains broadly applicable. Compared with ATP, WVS results register a higher per-survey cost at approximately \$1.73, due to the higher number of survey questions (185 for US and GB, 172 for HK). Notably, the US results are nearly identical across survey programs (ATP: 0.014; WVS: 0.015), providing evidence against program-specific overfitting (Table 8).

## D.3. Ablation Studies

### D.3.1. ATTRIBUTES

As discussed in Section 3.2, attributes play the role of control handles in P2P's generation logic, and *freeform* templates allow P2P to derive attributes from a specific survey or question, complementing the preset attribute bank with data-driven insights on endowment generation. To evaluation the potency of the freeform templates, we conduct an ablation in study in which we sequentially turn off question patching, survey patching and both. We additionally test the impact of per-endowment attribute cap through a setting where the default max attribute=10 is changed to 20.

Results in Table 9 suggest that performance of P2P is robust to attribute cap, whereas turning off freeform templates (especially question patching) degrades both average entropy and test MSE without significant cost gain. Therefore, from a performance vantage, it is recommended to keep the freeform templates for the extra benefits of data-driven insights.

### D.3.2. ENDOWMENT BUDGET

We run P2P for 9 different endowment budgets, scaled linearly from 130 to 450. We increase the update steps of active endowment generation accordingly to ensure that at each step the number of new endowments generated is fixed at 20. We

*Table 9.* Attribute Ablation. Reported are mean ± std over 3 repeated runs.

| Metric | Avg. Entropy ↑ | Test MSE ↓ (Lasso) | Test MSE ↓ (Enet) | Cost (USD) ↓ |
|---|---|---|---|---|
| Original | **0.4451** ± .0256 | **0.0095** ± .0024 | **0.0079** ± .0026 | 0.9708 ± .0036 |
| Survey patch off | 0.4378 ± .0134 | 0.0103 ± .0011 | 0.0102 ± .0002 | 0.9785 ± .0122 |
| Question patch off | 0.3989 ± .0203 | 0.0114 ± .0016 | 0.0109 ± .0015 | 0.9698 ± .0011 |
| Freeform off | 0.3953 ± .0097 | 0.0128 ± .0008 | 0.0123 ± .0012 | **0.9688** ± .0015 |
| max attributes = 20 | 0.4363 ± .0061 | 0.0124 ± .0046 | 0.0115 ± .0041 | 0.9880 ± .0167 |

also include a budget of 110 with no updates as an ablation of adaptive sampling. Table 10 shows that on W42, a higher budget generally improve test performance but gains plateau after about three updates, indicating a moderate budget suffices.

*Table 10.* Performance of P2P with varying endowment budget. The average question entropy, test MSE (Lasso and ElasticNet), and generation cost are repeated. All values are mean ± std over 3 repeated runs.

*(a)* Budgets 110–210

| Endowment Budget | 110 | 130 | 170 | 210 |
|---|---|---|---|---|
| Avg. Entropy ↑ | 0.3777 ± 0.0191 | 0.3697 ± 0.0057 | 0.4180 ± 0.0080 | 0.4341 ± 0.0043 |
| Test MSE (Lasso) ↓ | 0.0144 ± 0.0018 | 0.0124 ± 0.0043 | 0.0167 ± 0.0041 | 0.0100 ± 0.0025 |
| Test MSE (Enet) ↓ | 0.0149 ± 0.0014 | 0.0123 ± 0.0045 | 0.0156 ± 0.0028 | 0.0103 ± 0.0026 |
| Cost (USD) ↓ | 0.3487 ± 0.0035 | 0.8043 ± 0.0032 | **0.4108** ± 0.0089 | 0.5453 ± 0.0063 |

*(b)* Budgets 250–330

| Endowment Budget | 250 | 290 | 330 |
|---|---|---|---|
| Avg. Entropy ↑ | 0.4179 ± 0.0165 | 0.4369 ± 0.0168 | 0.4308 ± 0.0208 |
| Test MSE (Lasso) ↓ | 0.0111 ± 0.0006 | 0.0090 ± 0.0013 | 0.0133 ± 0.0030 |
| Test MSE (Enet) ↓ | 0.0104 ± 0.0006 | 0.0097 ± 0.0017 | 0.0121 ± 0.0031 |
| Cost (USD) ↓ | 0.6715 ± 0.0129 | 0.9365 ± 0.0027 | 1.0688 ± 0.0119 |

*(c)* Budgets 370–450

| Endowment Budget | 370 | 410 | 450 |
|---|---|---|---|
| Avg. Entropy ↑ | 0.4216 ± 0.0179 | **0.4540** ± 0.0142 | 0.4433 ± 0.0092 |
| Test MSE (Lasso) ↓ | 0.0140 ± 0.0037 | **0.0080** ± 0.0006 | 0.0099 ± 0.0020 |
| Test MSE (Enet) ↓ | 0.0136 ± 0.0055 | **0.0084** ± 0.0004 | 0.0097 ± 0.0020 |
| Cost (USD) ↓ | 1.2050 ± 0.0179 | 1.3295 ± 0.0142 | 1.4619 ± 0.0092 |

### D.3.3. MODEL BACKEND

We test P2P's performance on five model backends: GPT-4.1-mini, GPT-4.1-nano, Gemini-2.0-flash, Gemini-2.5-flash and a locally hosted Qwen. Performance varies substantially, with the best average question entropy (0.471) and test MSE (0.009) achieved by GPT-4.1-mini (Table 11). Gemini-2.0-flash ranks second on these metrics but is markedly more cost-efficient, requiring only one quarter of the cost of GPT-4.1-mini. Smaller models, like GPT-4.1-nano and Gemini-2.5-flash-lite, perform worse, consistent with our theoretical intuition that limited model capacity constrains the learnability condition (Definition 2.2).

### D.3.4. REGRESSION-BASED AGGREGATION

To gauge the contribution of the regression-based aggregation module, we ablate it by replacing regression with simple averaging of agent responses, which increases the test MSE from 0.0104 to 0.0254 (Table 12). In essence, regression not only distills the agent collection into a more compact ensemble, but it also plays a crucial role in ensuring the population

*Table 11.* Comparison of `P2P` performance on W42 using different model backends. Reported are average question entropy, test MSE (Lasso and ElasticNet), and generation cost. For each model, we fix the endowment budget at 300. All values are mean $\pm$ std over 3 repeated runs.

| Model | Avg. Entropy | Test MSE (Lasso) | Test MSE (Enet) | Cost (USD) |
|---|---|---|---|---|
| GPT-4.1-mini | **0.471** | **0.0090** | **0.0088** | 3.83 |
| | (0.017) | (0.0014) | (0.0014) | (0.045) |
| GPT-4.1-nano | 0.3996 | 0.0226 | 0.0213 | 0.9265 |
| | (0.0154) | (0.0042) | (0.0048) | (0.0043) |
| Qwen | 0.193 | 0.0204 | 0.0196 | – |
| | (0.040) | (0.0014) | (0.0027) | |
| Gemini-2.0-flash | 0.445 | 0.0104 | 0.0091 | **0.971** |
| | (0.026) | (0.0038) | (0.0046) | (0.004) |
| Gemini-2.5-flash-lite | 0.360 | 0.0149 | 0.0151 | 1.023 |
| | (0.0008) | (0.0009) | (0.0013) | (0.0010) |

*Note.* Qwen was run on a local server, thereby incurring no API cost.

preference reflected in the survey data are actually learned by the ensemble.

*Table 12.* Comparison of `P2P` performance with and without the regression-based aggregation stage on ATP W42. In the ablation, agent responses from active endowment generation are combined by simple averaging. For the regression setting, we use test MSE for Lasso.

| Metric | Without regression | With regression |
|---|---|---|
| Test MSE (Lasso) $\downarrow$ | $0.0254 \pm 0.00041$ | $\mathbf{0.0104} \pm 0.00383$ |

We also compare the default option-level weighting against block weighting, where each question contributes equally to the loss. Performance remains comparable under both schemes for W42 (Table 13).

*Table 13.* Sensitivity check of weighting schemes. Reported are mean $\pm$ std over 3 repeated runs.

| Weighting Scheme | Train MSE $\downarrow$ (Lasso) | Test MSE $\downarrow$ (Lasso) | Train MSE $\downarrow$ (Enet) | Test MSE $\downarrow$ (Enet) |
|---|---|---|---|---|
| Default | $0.00526 \pm .00141$ | $0.00946 \pm .00241$ | $\mathbf{0.00353} \pm \mathbf{.00161}$ | $\mathbf{0.00792} \pm \mathbf{.00261}$ |
| Block | $\mathbf{0.00472} \pm \mathbf{.00228}$ | $\mathbf{0.00884} \pm \mathbf{.00281}$ | $0.00389 \pm .00152$ | $0.00797 \pm .00263$ |

### D.4. Stress Test: Comparison with an SFT-Aligned Model under Topic Shift

`P2P` is designed as a plug-and-play preference alignment method that can be deployed on new populations with limited-data and limited-compute regimes. In the main paper, the baseline comparisons thus focus on prompting-based ensemble methods under the same constraints. As discussed in Appendix E, supervised finetuning (SFT) for a distribution-calibrated model offers an alternative approach under more lenient data and compute environments. An exemplar is Cao et al. (2025), where SFT is used to train a predictive model based on pooled World Values Survey (WVS) data from 46 countries, and conditioned on locale to directly predict answer distribution for a given survey question. They further test the SFT-aligned model on a distinct unseen set of WVS questions along with other cross-country and cross-dataset validation tasks.

In this regard, our work differs from Cao et al. (2025) not just in terms of training regimes, but also in key methodological premises. Specifically, we regard each survey panel as a benchmark for *a specific population preference on a given topic at a given moment in time*, where such preference is *gaugeable* through survey responses (cf. Figure 1, right panel). We do *not* assume that preferences on a topic domain can be reliably inferred if the training questions never probe that domain. For example, inferring political attitudes solely from non-political or religious items is a strong assumption for both humans and synthetic agents. Likewise, we do *not* assume preferences can carry over from one survey program to another, particularly

when the surveys are conducted at different points in time and on different subpopulations of the same locale. Studying how aligned preferences generalize across topics, survey programs and over time is itself a key direction for future work, and will require both theoretical and empirical advances.

Nevertheless, Cao et al. (2025)'s setup constitutes an interesting stress test for P2P outside our studied scenarios. We present in this section a head-to-head comparison with their method on their WVS arena. We follow their protocol: Q1 and Q2 for train/validation and Q3 as test, which induces a topic shift from general attitudinal & religious/ethical items (Q1+Q2) to political interest and culture items (Q3). In our language, this asks: if we align to a population on broad attitudinal dimensions (Q1+Q2), how well does the aligned ensemble generalize to a new unseen dimension (Q3: political-cultural questions)? As in Section D.2, we focus on the three representative locales: US (overlaps ATP), Great Britain (Western Europe) and Hong Kong SAR (East Asia). For each, we filter Cao et al. (2025)'s data to that locale, train P2P (backend: gemini-2.0-flash; 300 endowments) on local Q1+Q2 (train+val), and evaluate on Q3. Table 14 shows the training schemes for P2P and Cao et al. (2025).

*Table 14.* Different training regimes for P2P and (Cao et al., 2025).

| Setup | Train+Val Entries (Q1 + Q2) | Test Entries (Q3) | Model backend |
|---|---|---|---|
| P2P (US) | 185 (150 + 35) | 59 | gemini-2.0-flash |
| P2P (GB) | 172 (140 + 32) | 60 | gemini-2.0-flash |
| P2P (HK) | 185 (150 + 35) | 59 | gemini-2.0-flash |
| Cao et al. (C1: 46 countries) | 8427 (6841 + 1586) | 2719 | Llama3-8B-Instruct |

*Table 15.* P2P vs. Cao et al. (2025) on WVS (Q3). Reported are mean $\pm$ std over 3 repeated runs.

| Setup | Method | Test MSE $\downarrow$ | Test 1-JSD $\uparrow$ | Test MCPA $\uparrow$ | Test EMD $\downarrow$ |
|---|---|---|---|---|---|
| P2P (US) | Lasso | $0.0217 \pm .0014$ | $0.7492 \pm .0036$ | $0.6271 \pm .0143$ | $0.1188 \pm .0048$ |
| P2P (US) | Enet | $0.0218 \pm .0019$ | $0.7497 \pm .0051$ | $0.6384 \pm .0317$ | $0.1184 \pm .0066$ |
| P2P (GB) | Lasso | $0.0249 \pm .0009$ | $0.7416 \pm .0055$ | $0.4889 \pm .0347$ | $0.1296 \pm .0045$ |
| P2P (GB) | Enet | $0.0249 \pm .0009$ | $0.7463 \pm .0067$ | $0.5111 \pm .0096$ | $0.1294 \pm .0048$ |
| P2P (HK) | Lasso | $0.0284 \pm .0023$ | $0.7115 \pm .0125$ | $0.4802 \pm .0353$ | $0.1507 \pm .0122$ |
| P2P (HK) | Enet | $0.0290 \pm .0023$ | $0.7151 \pm .0133$ | $0.4689 \pm .0298$ | $0.1532 \pm .0108$ |
| Cao et al. | SFT | $-$ | $0.777$ | $0.43$ | $-$[*] |

[*] EMD not reported due to incompatible definition in Cao et al. (2025).

Table 15 presents the results on the performance comparison. In summary, P2P achieves 1-JSD on Q3 comparable to Cao et al. (2025)'s SFT model and matches or exceeds their majority-class prediction accuracy (MCPA), despite using far fewer training entries per locale and no fine-tuning (Table 14). Following our earlier discussion, we interpret performance on Q3 as evidence that some preference structure transfers across topics, rather than as a requirement that a preference-alignment method must perfectly extrapolate to arbitrary unseen domains.

## E. Extended Literature Review

### E.1. Pluralistic Alignment

AI alignment refers to the process of ensuring that an AI system operates in accordance with human intentions and values, whether at an individual or aggregate level (Ji et al., 2023; Leike et al., 2018). As AI systems are adopted by increasingly diverse users, they must be designed to recognize and address a wider range of needs. This necessitates for pluralistic systems capable of capturing and representing the diversity of human values and perspectives (Sorensen et al., 2024b). Pluralistic alignment therefore seeks to align models not with a single gold standard, but with a diverse range of user preferences across various attributes.

Most works on pluralistic alignment touch on three complementary ways, namely overton, steerable, and distributional pluralism, in which a single AI model or system can support diversity of views. In overton pluralism, the model aims to output a whole spectrum of reasonable responses (Sorensen et al., 2024a). Methods to achieve this include looking at alignment shifts to transform distributions toward overton coverage (Lake et al., 2024) or few shot prompting from community-specialized LLMs (Feng et al., 2024). In steerable pluralism, models are typically steered to adopt or favor particular perspectives or value attributes, often through few-shot examples (Feng et al., 2024; Adams et al., 2025). By contrast, our approach steers purely through attributes, with the goal of constructing diverse agents that embody different perspectives. In distributional pluralism, the distribution of the model over possible answers is intended to match that of some target population (Sorensen et al., 2024b), often achieved through model fine-tuning on human responses (Cao et al., 2025).

Another way to categorize pluralistic alignment work is by the amount of data they rely on. While most approached in pluralistic alignment involve post-training, they range from resource-intensive approaches that fine-tune on large annotated corpora to lightweight methods that operate with minimal or no additional data. With sufficient data, specialized community models can be fine-tuned and subsequently combined using model merging techniques (Yuan et al., 2024), few-shot prompting from community model outputs (Feng et al., 2024), or federated averaging (Srewa et al., 2025) to scale with user diversity. If limited data are available, an alternative line of post-training approaches leverage in-context learning, where similarity-based retrieval (Adams et al., 2025) or group-informed retrieval (Chen et al., 2024) is performed to select few-shot examples that guide models toward pluralistic alignment. In the absence of detailed data from individual pluralistic users, a third post-training approach is inference-time conditioning. Inference-time conditioning uses structured prompts that encode sociodemographic traits or behavioral dispositions to condition an LLM to emulate the responses of a specific agent. These agents are also known as *endowments*, personas (Castricato et al., 2025) or Silicon Samples (Horton, 2023; Argyle et al., 2023) and are lightweight to deploy and data-efficient. As inference-time conditioning is cheap and fast, they enable mixture or vector-based formulations of pluralistic alignment, which combine agents via multi-objective rewards to avoid averaging out minority perspectives (Chen et al., 2025; Feng et al., 2025). Therefore, `P2P` follows this line of work for pluralistic alignment. In the first step, `P2P` performs inference-time conditioning to generate a sufficient set of agents that adequately span the preference space. Thereafter, in the second step, it combines agents to achieve a compact yet pluralistic set of preferences that represent the target survey population.

#### E.1.1. ATTRIBUTES

Pluralistic alignment works that use inference-time conditioning often steer agents toward specific profiles using *attributes* that include moral and value dimensions (Adams et al., 2025), or broad morals, values, characteristics, and perspectives (Sorensen et al., 2024a). Attributes may also refer to demographic features, that are more tangible. Examples of these attributes include age, sex, education, income, and religion (Castricato et al., 2025). `P2P` uses the term *attributes* more broadly to encompass both moral and value dimensions and demographic features. Bundling these together allows `P2P` to define and condition LLM agents in a way that reflects both who they are and how they evaluate options.

Other related works on pluralistic alignment relate to the curation of datasets, which support research on pluralistic alignment (Sorensen et al., 2024a; Zhou et al., 2025).

### E.2. Large Language Models to Emulate Human Preferences

Leveraging the ability of LLMs to emulate human or human-group responses with considerable fidelity, LLM alignment or personalization (Tseng et al., 2024) has been largely focused on emulating human responses using various techniques. These include learning from feedback or reward models (Bai et al., 2022; Christiano et al., 2017), under distribution shifts (Leike

et al., 2018; Krueger et al., 2021), finetuning (Tan et al., 2024), retrieval augmented generation (Prahlad et al., 2025), representation learning (Ren et al., 2024), and using human-generated (Horton, 2023; Castricato et al., 2025; Argyle et al., 2023) prompts. However, when diverse responses need to be generated from many users or user groups instead of individuals, the data and time resources required for alignment grow linearly with the number of LLMs that need to be aligned. To reduce alignment costs, some works instead rely on AI-generated (Schuller et al., 2024; Salminen et al., 2024; Ge et al., 2025; Simmons, 2023) prompts, or alignment to multiple user groups (Mondal et al., 2025; Zhao et al., 2024b), even entire populations (Cao et al., 2025). However, even the best models struggle with this task (Cao et al., 2025). Furthermore, responses from LLMs have been found to have less variation than responses from real surveys (Bisbee et al., 2024), and LLMs have a tendency to respond in the middle category (Wang et al., 2024). Consequently, it is imperative to address these limitations and devise data- and time-efficient approaches that enable LLMs to more accurately and reliably serve as proxies for survey populations. Our work contributes to this goal by introducing an active learning–inspired alignment framework that dynamically constructs a diverse agent basis under resource constraints and reconstructs population-level preferences via regression-based aggregation, without relying on ground-truth demographic profiles.

### E.3. Data Selection and Active Learning

To increase preference coverage while operating under budget constraints, our active endowment generation pipeline draws inspiration from machine learning paradigms, particularly active learning.

Traditional data selection methods aim to improve the data efficiency of supervised deep learning models by identifying the most informative or important training examples for generalization (Coleman et al., 2020; Paul et al., 2021). Selection via Proxy (SVP) (Coleman et al., 2020) uses a lightweight proxy model to estimate sample utility at a fraction of the cost, while GraNd and EL2N scores (Paul et al., 2021) identify important training examples early in training.

Similarly, deep active learning applies query strategies to iteratively select the most informative samples from a large pool of data to be added to the training dataset for retraining or fine-tuning (Astorga et al., 2024; Hübotter et al., 2025; Li et al., 2025). Active learning strategies balance exploitation, focusing on uncertain or high-impact samples, with exploration, seeking diverse and novel samples to broaden coverage.

In our system, we apply this logic to guide both endowment generation and question patching. Specifically, low-entropy responses are treated as indicators of low coverage or model uncertainty. We refer to question patching as a boosting strategy—not in the ensemble learning sense, but to distinguish it from the main adaptive sampling process—since it targets underperforming questions by allocating additional sampling to enhance expressivity. Meanwhile, the exploitation–exploration mechanism used to sample across existing modes reflects the same active learning principle, but with a different emphasis: prioritizing high-utility regions while maintaining representational diversity. Together, these components form an active learning–inspired loop that incrementally expands coverage of the preference space while minimizing redundant agent generation.

## F. Limitations and Future Work

**Question weighting and regression loss**   By design, `P2P` converts all multiple choice questions into binary questions before fitting a regression model. For example, a question with five options will turn into five questions with binary options. While this design ensures that regression can run smoothly with heterogeneous question types, it steers the model to prioritize fitting questions with more answer options, e.g., a 10-point Likert scale. Put differently, information contained in questions with fewer answer options, e.g., Yes/No, gets diluted after the binarization. Whether or not this design is desirable depends on the user's assumption of the relationship between preference signal and choice granularity. The current design implicitly posits a positive relationship between the two—i.e., the finer the scale, the better a signal on the latent preference. For other assumptions, question reweighting or format transformation need to be done before the regression step. `P2P` uses option-level weighting as the default, but also offers block weighting as an option.

**Survey data in, text out**   The current design of `P2P` limits its use case to labeling: Given question and options, the aligned agent ensemble expresses its preference by offering a probability distribution over the given options. A key future research direction is to extend the use case to text completion, e.g., drafting resolution, offering suggestion and answering freeform survey questions. How to conflate individual agent textual responses into a representative aggregate response remains an under-explored area of research. Auction design (cf. Dütting et al., 2024) can be a promising aggregation algorithm in this case.

**P2P as a benchmark for model steerability**   In this study our primary goal is understanding *preference reconstruction theory* and its implementation through the two-stage alignment framework. While our focus is on the design aspects of the system, we acknowledge that the backend model's steerability is also a determinant of `P2P`'s empirical performance. The ablation study using different backends offers first insights on this aspect. It also indicates `P2P`'s potential of being transformed into a benchmark pipeline for the study of model steerability in pluralistic alignment.

**Temporal Alignment and Non-stationary Preference**   Any method trained at a single moment risks becoming stale as attitudes and beliefs shift. This also affects `P2P` when considering using a fitted ensemble in future surveys. Our view is that synthetic agents should be periodically re-anchored using more recent, easier-to-collect data (e.g., shorter or cheaper surveys) and then tested on more demanding items. Architecturally, `P2P` is set up for this. The endowment generation and regression stages can be rerun on new waves, and one can track how learned weights and entropy patterns evolve, or when reconstruction quality begins to deteriorate. While `P2P` provides the machinery, systematic studies of preference shift and alignment across time and topics remain open.

**Short-cut Learning and Generalization**   Overfitting to survey artifacts is a key concern for *any* survey-based evaluation. This concern is closely related to what the robustness literature calls *shortcut learning* or *Clever Hans* behavior, where models exploit superficial regularities of a dataset instead of learning the underlying concept (Geirhos et al., 2020; Lapuschkin et al., 2019). Our empirical evaluation across 14 ATP waves is designed to mitigate the risk that our findings hinge on idiosyncrasies of a single wave, and the additional WVS experiments further mitigate the risk of ATP-specific shortcutting. While these steps serve as initial evidence against program-specific shortcutting, it does not constitute a complete solution to the generalization problem. A full multi-survey study (e.g., across GlobalOpinionQA and other polling corpora) is beyond the scope of this paper, but `P2P`'s modular pipeline readily supports this kind of analysis. Fundamentally, we see this as partly a survey design challenge: if we want synthetic agents to learn stable preferences rather than survey-specific shortcuts, future work may need survey instruments that are explicitly constructed with learnability and cross-topic generalization in mind. In the last paragraph of Appendix , we suggest a semantic question module for `P2P` to quantify the expressiveness of survey questions, which could serve as a parallel line of work to mitigate shortcutting on the model-design side. Taken together, these directions highlight an important and promising interdisciplinary research frontier in our pursuit of alignment science.

