# OpenReview forum: "Prompts to Proxies: Emulating Human Preferences via a Compact LLM Ensemble"
_ICML.cc/2026/Conference — Submitted to ICML 2026_

### Official Review · Reviewer_aZG6 · 2026-03-09

**Soundness:** 2
**Presentation:** 3
**Significance:** 2
**Originality:** 2
**Overall Recommendation:** 4
**Confidence:** 3

**Summary:**

This paper studies the problem of using synthetic agents to faithfully represent the preferences of a target human population. The authors formulate preference alignment as a representation learning problem by constructing a functional basis of proxy agents and recovering population preferences through a weighted aggregation of these agents. To implement this idea, the paper proposes a two-stage system called Prompts to Proxies (P2P). In the first stage, entropy-based adaptive sampling is used to construct a diverse pool of agents. In the second stage, L1 regularized regression is applied to select a compact ensemble of proxy agents.

**Compliance With Llm Reviewing Policy:**

Affirmed.

**Final Justification:**

The authors clarified the efficiency of the adaptive sampling strategy based on entropy in their rebuttal and demonstrated that rating tasks can be seamlessly integrated into their framework. These clarifications largely resolve my prior concerns and I have accordingly raised my score.

**Key Questions For Authors:**

1. In Lines 256 to 267, how are the parameters $k$ and $t$ determined in practice?

**Limitations:**

Yes.

**Strengths And Weaknesses:**

Strengths:
1. The paper proposes a two-stage system called Prompts to Proxies (P2P). In the first stage, entropy-based adaptive sampling is used to construct a diverse pool of agents. In the second stage, L1 regularized regression is applied to select a compact ensemble of proxy agents.
2. The paper includes extensive empirical evaluations that support the effectiveness of the proposed method.
---
Weaknesses:
1. The paper proposes an entropy-based adaptive sampling strategy to generate proxy prompts and argues that this improves diversity and coverage of the preference space. However, it is not entirely clear why maximizing response entropy necessarily leads to better coverage of latent preference types. High entropy may also arise from noise or poorly calibrated proxy responses rather than meaningful disagreement across preference groups. Some additional justification or analysis would strengthen this argument.
2. The paper suggests that the entropy-based sampling strategy allows the proxy ensemble to approximate the population preference distribution. However, it is not clear whether there are theoretical guarantees that the resulting proxy mixture converges to the true population distribution.
3. The experiments mainly focus on multiple-choice or categorical response settings, where entropy over response options is well defined. It is less clear how the proposed framework extends to other types of preference data, such as rating tasks or open-ended text generation tasks.

---

> ### Author Rebuttal · Authors · 2026-03-28
>
> We thank the reviewer for their careful reading and evaluation of P2P as a two-stage system. Fundamentally, P2P marks our effort to instantiate the proposed preference reconstruction theory—a new paradigm for preference alignment motivated by formal expressive gaps and identification issue of existing paradigms (Observations 2.4 and 2.5). Grounded in representation learning and statistical inference, this approach achieves comparable or better fidelity with notably lower resource demands (no finetuning/demographics). We elaborate on the paradigmatic contribution in our response to Reviewer CFwr’s W1 and W2, and address the reviewer’s concerns in order:
>
> **[W1]** The reviewer’s observation that noise could possibly contribute to entropy is spot-on. Figure 1 affords an intuitive understanding of the relationship. Essentially, entropy, or response diversity in general, is an indirect gauge of the latent preference diversity. It is best viewed in an econometric sense, where the relationship is correlational rather than causal. Optimizing entropy is an efficient expedient when direct observation of preference is unavailable. As the reviewer rightly notes, other factors (like irrelevant model idiosyncrasies or unrelated preference dimensions) can also lead to response differences—this reflects the many-to-one mapping from preference to response. This is fundamentally an identification issue not related to P2P alone but affecting data-driven learning paradigms in general. In P2P, we guard against overfitting or noise fitting with the validation questions, which are used to select the hyperparameter(s) at the aggregation stage. Simulation Study 3 in Appendix C.3 provides empirical evidence corroborating the use of entropy as the Stage 1 objective. We refer to lines 1541 to 1557, which directly address the reviewer’s concern on noise fitting.
>
> **[W2]** Formalizing convergence guarantees for preference reconstruction is non-trivial, for proxy agents are actively constructed rather than drawn from a fixed distribution, placing this outside the scope of standard statistical convergence results. We appreciate the reviewer flagging this as a limitation. What we do provide is (1) a learnability framework (Definition 2.2, Observations 2.3-2.5) characterizing the feasibility of reconstruction in principle; (2) simulation studies with known ground truth indicating reconstruction error decreases with proxy pool expressiveness (Appendices C.2 and C.3); (3) empirical experiments across ATP and WVS demonstrating consistent real-world performance, and ATP-based ablation showing performance stabilizes after 3 update steps (Appendix D.3.2). Developing error bounds (how target reconstruction error depends on proxy pool expressiveness, calibration-target question shift, and finite sample size) is an active direction in our theoretical sequel. We will flag this more explicitly as a limitation and future work in the revision.
>
> **[W3]** Rating tasks (E.g., Likert scales and AB-testing pairs) are already within scope. P2P’s binarization step converts any ordinal scales or nominal categorization into option-level probabilities (Equations 6 and 7), and our evaluation on ATP and WVS contains questions with varying scale granularity. In fact, the pipeline design readily supports any data that can be structured in categorical or discretized formats, not only surveys. Our focus on survey data is motivated by their public availability, transparent documentation, and well-defined target populations (Section 1, lines 059-064). When textual data can be converted or reduced to a set of categorical signals (e.g., sentiment, stance, tone), application of P2P is standard. The deeper challenge, which we discuss in Appendix F, arises when the goal is to aggregate textual responses into a unified policy recommendation. This requires innovation in text-level aggregation that goes beyond weight estimation. We have not incorporated text-to-categorical preprocessing into P2P precisely because standard well-established social surveys, like ATP and WVS, contain no free-text entries, and hence no ground truth to validate such a module against. Curation of open-ended survey datasets with well-defined populations and clear documentation is an important complementary direction for future research.
>
> **[Q1]** In all our main experiments, we used default values of 3 for $k$ (number of lowest entropy questions to be patched per round) and 2 for $t$ (threshold for triggering mixed-mode strategy), without specific hyperparameter search. Ablation on templates (Appendix D.3.1) indicates turning off question patch (effectively setting $k$ to 0) degrades entropy and increases test MSE (Table 9), and Figure 9 offers a snapshot of how different templates contribute non-trivially to the final selected ensemble. In practice, optimal values for $k$ and $t$ likely depends on the survey and topic domain, so we leave systematic tuning guidance to application-specific studies.

---

> > ### Author Rebuttal · Reviewer_aZG6 · 2026-04-03
> >
> > Thanks for the detailed response. I decide to increase the score to 4.

---

> > > ### Author Response · Authors · 2026-04-05
> > >
> > > We thank the reviewer for their acknowledgement on full resolution of concerns, and for deciding to raise their score to 4.

---

### Official Review · Reviewer_bnL8 · 2026-03-11

**Soundness:** 3
**Presentation:** 3
**Significance:** 2
**Originality:** 2
**Overall Recommendation:** 4
**Confidence:** 2

**Summary:**

This paper studies whether LLMs can serve as faithful proxies for human populations in social science settings. It introduces preference reconstruction theory, which frames
  population preference alignment as a representation-learning problem: instead of fitting a single representative agent, the goal is to construct a functional basis of proxy agents
  and recover population-level preferences through weighted aggregation. Based on this idea, the paper proposes Prompts to Proxies (P2P), a two-stage inference-time system. In Stage 1,
  it actively generates a diverse pool of persona-like agents using structured attributes, entropy-based adaptive sampling, and question patching. In Stage 2, it uses L1-regularized
  regression to select a compact ensemble whose aggregated response distribution matches survey data. Empirically, the paper evaluates P2P on 14 waves of the American Trends Panel and
  on the World Values Survey, showing lower test MSE than prompting baselines and competitive performance against a distribution-calibrated SFT model while using much less data and no
  finetuning.

**Compliance With Llm Reviewing Policy:**

Affirmed.

**Final Justification:**

The authors clarified the scope and positioning of their claims, strengthened the justification for their empirical design and baseline choices, and provided ablations showing that both adaptive endowment generation and regression are important to performance. These clarifications largely address my prior concerns, so I am maintaining my positive score.

**Key Questions For Authors:**

see weaknesses

**Limitations:**

yes

**Strengths And Weaknesses:**

# Strengths

- This paper addresses an important and timely problem. Using LLMs as proxies for human populations is increasingly common, but external validity remains a central concern.
- Casting pluralistic preference alignment as a reconstruction problem over a functional basis is conceptually interesting and more distinctive than a purely engineering-oriented prompting paper.
- Evaluating across 14 ATP waves, three WVS locales, ablations, transfer/native analysis, and a stress test against an SFT-based approach gives the paper more credibility than a single-dataset demonstration.
- The method is appealing for low-resource settings. It avoids finetuning, does not require demographic data, and appears inexpensive to run.

#  Weaknesses

- The strongest concern is whether the task is partly easier than the paper’s framing suggests. The method predicts aggregate survey response distributions over fixed answer choices, not open-ended human preferences in a broader sense. This is still useful, but the claims should be calibrated carefully.
- The main empirical baselines are somewhat limited. The two direct baselines are prompting-based ensemble methods, and while the paper includes a stress test against an SFT-aligned model, the core comparison would be stronger with more competitive pluralistic-alignment or survey-emulation baselines.
- The theoretical component is interesting but still somewhat aspirational. The representation-learning framing helps motivate the system, but the connection between the theory and the concrete design choices in P2P is not always tight. In particular, the entropy objective and adaptive sampling strategy remain fairly heuristic.
- The regression stage may be doing a substantial amount of the work. The paper does show that simple averaging performs worse, but it remains somewhat unclear how much of the gain comes from Stage 1’s sophisticated proxy construction versus Stage 2’s sparse regression fitting.

---

> ### Author Rebuttal · Authors · 2026-03-28
>
> We thank the reviewer for their valuable feedback, and address their concerns in order:
>
> **[W1]** We appreciate this nuance and agree that claims should be calibrated. For P2P, we choose to focus on categorical survey responses as they provide the cleanest and most standard ground truth for population preference alignment. As noted in lines 059-064, social survey data are publicly available, transparently documented, and designed to represent well-defined populations. We view this as the natural first testbed for population preference reconstruction. We note that despite the seemingly constrained format, existing methods do not address this task well (Table 1), and also target it with heavier arsenal (e.g., Cao et al. 2025’s full SFT on thousands of training samples). A contribution of P2P is establishing that a reconstruction approach rooted in first principles can match such resource-heavy methods, which we see as the necessary foundation before scaling to richer response formats and motivation for further study in this direction. The performance gap (Table 2) and error analysis (Section 4.4) show the problem has genuine structure. The preference reconstruction theory itself (Section 2) is stated for abstract preferences and responses, thus not confined to categorical data. Extension to open-ended text generation is discussed as a key future direction in Appendix F, where the main challenge is aggregation and evaluation of free-form textual responses. We will review the manuscript to ensure that the framing accurately reflects the current empirical scope.
>
> **[W2]** We share the reviewer’s interest in broader comparisons. In fact, the stress test against Cao et al. (2025) in Section 4.5 is a direct comparison against the most comparable SFT-based survey-emulation method we are aware of. As noted in our response to Reviewer CFwr's W5, we additionally report clustering and raw model backend baselines (Appendix D.1.1 and D.1.2), which do not outperform our main baselines. On pluralistic alignment methods, existing approaches in this space (e.g., Feng et al. 2024, Chen et al. 2025, Adams et al. 2025) differ significantly from P2P in both data requirements and objectives, which prevents a meaningful head-to-head—we discuss these distinctions in Appendix E.1. For the empirical baselines reported in the main paper, Vanilla is designed to isolate the effect of P2P’s active endowment generation, whereas PERSONA marks our best effort to benchmark against a demographic conditioning baseline. In addition, we note that Park et al. (2024), whose generative agent simulations of 1000 people would constitute a natural demographic conditioning baseline and a downstream application testbed, has not yet released their data. We plan to incorporate this comparison when the data becomes available. We would welcome the suggestion if the reviewer has a specific pluralistic-alignment or survey-emulation method in mind that operates on aggregate survey data without individual-level supervision.
>
> **[W3]** We appreciate this distinction and agree it is important. As noted in our response to Reviewer CFwr’s W2, we deliberately make the theoretical formulation in Section 2 conceptual and technique-agnostic to expose the structural relationships between the existing alignment paradigms and our preference reconstruction approach. As the reviewer rightly notes, subsequent design choices in P2P concretize the theory but lack tailored theoretical guarantees, which is an active direction in our follow-up work. Optimizing the entropy objective through adaptive sampling is our operationalization of theoretical requirement to construct a diverse set of proxy agents, which we validate empirically. Simulation Study 3 (Appendix C.3) indicates higher entropy coverage generally leads to better reconstruction performance, and ablation on updating steps (Appendix D.3.2) and attribute templates (Appendix D.3.1) confirm adaptive sampling strategy’s contribution. The gap between the theoretical principle of maximizing coverage and the implementation of optimizing entropy is bridged empirically through these results.
>
> **[W4]** Our current experiments and ablation studies are designed to address the reviewer’s question about the relative importance of components. Table 12 (Appendix D) shows removing regression while keeping P2P’s active endowment generation worsens test MSE from 0.010 to 0.025. In contrast, replacing the entropy-based endowment generation with Vanilla generation degrades test MSE from 0.014 to 0.026 (Table 1). This confirms that both components matter in the overall performance. Ablation studies on finer-grained subcomponents (attributes, endowment budget and model backend) are provided in Appendix D.3. We note that both Reviewers bnL8 and CFwr raised this question, which suggests the relative contributions of each stage are not sufficiently foregrounded in the current manuscript. We will make this comparison more prominent in the revision.

---

> > ### Author Rebuttal · Reviewer_bnL8 · 2026-04-03
> >
> > I have carefully reviewed all the rebuttal responses. I decide to maintain my current score.

---

> > > ### Author Response · Authors · 2026-04-05
> > >
> > > We thank the reviewer for their acknowledgement and we are glad that our rebuttal responses fully resolve their concerns.

---

### Official Review · Reviewer_CFwr · 2026-03-15

**Soundness:** 3
**Presentation:** 3
**Significance:** 2
**Originality:** 2
**Overall Recommendation:** 3
**Confidence:** 4

**Summary:**

This paper studies the problem of aligning large language models with population-level human preferences. Specifically, the authors propose a framework named Prompts to Proxies (P2P), which models preference alignment as a representation learning problem. The key idea is to construct a diverse pool of proxy agents via prompt engineering and entropy-based adaptive sampling, and then select a compact ensemble of agents using L1-regularized regression so that their aggregated responses match the observed survey responses of the target population. The experiments on the American Trends Panel and World Values Survey demonstrate the effectiveness of the proposed method.

**Compliance With Llm Reviewing Policy:**

Affirmed.

**Key Questions For Authors:**

W1. The novelty and technical contribution appear somewhat limited.
The proposed framework mainly combines prompt-based persona generation with a regression-based aggregation method. However, generating diverse personas through prompt engineering and aggregating them via linear models (e.g., Lasso or Elastic Net) are well-known techniques in the literature. The reviewer feels that the technical innovation of combining these components is relatively limited.

W2. The theoretical formulation is not fully convincing.
Although the authors introduce the concept of “preference reconstruction theory” and formalize population preference as a weighted combination of proxy agents, the formulation mainly reduces to a regression-based ensemble model. The theoretical insights provided in the paper appear to be relatively straightforward adaptations of existing representation learning and aggregation frameworks.

W3. The experimental analysis could be strengthened.
The experiments mainly focus on survey prediction tasks. However, it is not entirely clear whether the improvements come from the entropy-based agent generation strategy or from the regression-based aggregation step. The authors should conduct more ablation studies to isolate the impact of each component in the framework.

W4. The scalability and computational cost are not fully discussed.
The method requires generating a large pool of proxy agents and evaluating their responses across many survey questions. It would be helpful if the authors could provide a more detailed analysis of the time complexity and runtime compared to other alignment approaches.

W5. The comparison with stronger baselines is limited.
The paper mainly compares with prompting-based ensemble baselines under constrained settings. However, it would be more convincing to include comparisons with other alignment approaches such as supervised fine-tuning or preference-learning methods, especially under comparable data regimes.

**Limitations:**

Yes.

**Strengths And Weaknesses:**

S1. The paper is generally well-written and easy to follow. The motivation of aligning LLM outputs with population-level preferences using survey data is clearly introduced, and the paper provides a detailed description of the proposed framework.

S2. The problem of modeling population-level preference distribution is interesting. The idea of representing a population preference as a weighted ensemble of proxy agents is intuitive and offers a new perspective on pluralistic alignment.

S3. The proposed method works reasonably well. Experimental results on several survey waves show that the proposed method consistently outperforms prompting-based baselines in terms of distributional metrics such as MSE, Jensen–Shannon similarity, and Earth Mover’s Distance

---

> ### Author Rebuttal · Authors · 2026-03-28
>
> We thank the reviewer for their constructive feedback, and address their concerns in order:
>
> **[W1]** The main novelty of our paper is paradigmatic. While existing preference alignment paradigms center around building “digital twins” (demographic conditioning) or a large one-size-fits-all model (SFT/RLHF), we offer a new perspective grounded in representation learning and statistical inference, showing it achieves comparable or better fidelity with substantially lower resource requirements—no finetuning, no profiling. As a zero-demographic method, P2P also serves a unique diagnostic role, affording a falsifiability check for claims that finer conditioning improves emulation. By requiring only API inference, publicly available survey data and a CPU to run, P2P lowers the barrier to entry for preference alignment research, enabling researchers without access to GPU clusters or privileged individual-level datasets to contribute to pluralistic alignment.
>
> On the technical side, we agree that persona prompting and regularized regressions are individually established techniques. What is new is their principled combination for operationalizing preference reconstruction. Our technical innovation lies in tailoring and enriching the techniques for this purpose (e.g., persona generation using extensible attribute banks, bandit-spirited question patching for human-LLM co-discovery of useful personas, encoding persona-driven agents so that they become selectable by regression algorithms).
>
> **[W2]** We wish to clarify a potential misread of the theoretical insights as straightforward adaptation of existing frameworks that reduce to a regression-based ensemble model. As with W1, each concept (learning, representation, aggregation) is long-standing in the literature; their contextualization in preference alignment is precisely what produces novel insights. Our theoretical formulation exposes failure modes of existing alignment paradigms (training a representative model via SFT/RLHF and demographic conditioning). In particular, Obs. 2.4 identifies a formal expressive gap, i.e. learnable preferences that neither representative agent nor demographic conditioning can recover. Obs. 2.5 further reveals an identification issue in demographic conditioning. Preference reconstruction theory is technique-agnostic, while the implementation via endowment generation and regression is a P2P design choice, motivated by the resource asymmetry observed in the intro (lines 030-037), though richer alternatives (e.g., training proxies via SFT and aggregating at the output layers) are equally valid. P2P is the first instance of this theory, and the comparisons with PERSONA (demographic conditioning) and Cao et al. (SFT) offer initial evidence on the new paradigm’s promise.
>
> **[W3]** Ablation evidence indicates both components contribute non-trivially. Ablating regression raises test MSE from 0.010 to 0.025 (Table 12), and replacing entropy-based generation with Vanilla lifts it from 0.014 to 0.026 (Table 1). We elaborate further in our response to Reviewer bnL8’s W4. Finer-grained ablations are in Appendix D.3.
>
> **[W4]** Except for the local server backend experiment, all experiments were run on a laptop with standard CPU and memory. Runtime is dominated by API inference. By design, P2P parallelizes both prompt generation and survey response elicitation, but the exact runtime varies by update_steps, max_workers, API rate limits and connection stability. For reference, a typical run on a single ATP wave (20 workers, 5 update steps, Gemini-2.0-Flash) takes about 26 mins, with regression under 2 minutes. Ablation (Table 10) shows performance plateaus after a pool size of 210 for ATP W42, suggesting that moderate pool sizes suffice. Total monetary cost is reported in the main tables and averages $0.8 per survey.
>
> **[W5]** We agree that comparison beyond prompting-based baselines is important. In Section 4.5, we compare P2P against Cao et al. (2025), a recent SFT-based method for survey response simulation for global populations. Despite differing premises (Appendix D.4), this is, to our knowledge, the closest comparison in the literature. Despite using less than 3% of the training data, P2P achieves comparable 1-JSD, and matches or exceeds their MCPA across the US, UK and HK locales. In Appendix D.1.1 and D.1.2, we additionally report clustering and raw model backend baselines, which do not outperform our standard baselines. Regarding comparison under “comparable data regimes,” the asymmetry is precisely the point. SFT methods require thousands of training entries, while P2P operates with a hundred or fewer training questions and API inference only. The practical contribution is achieving competitive fidelity under fundamentally lighter constraints. We welcome any suggestions of preference-learning methods that operate on aggregate survey response without individual-level data, and would be happy to discuss their applicability to our setting.

---

> > ### Author Rebuttal · Reviewer_CFwr · 2026-04-03
> >
> > I still feel the novelty and technical contribution appear somewhat limited. I decide to maintain my current score.

---

> > > ### Author Response · Authors · 2026-04-05
> > >
> > > We thank the reviewer for the acknowledgement. Since the remaining concern centers on novelty and technical contribution [W1], we offer the following observations.
> > >
> > > Firstly, to corroborate the paradigmatic nature of our contribution, we highlight a result that our preference reconstruction framework uniquely produces. The existing paradigms in preference alignment assume that faithful emulation requires matching the demographic or locale identity of the target population, be it through demographic conditioning or locale-conditioned SFT. In Section 4.4, we show this assumption is empirically falsified: **a more expressive basis originally generated for the US (US-native), after only Stage 2 weight refit (HK-transfer), outperforms HK-native generation on average (Figure 3B), demonstrating that basis expressiveness, not demographic matching, is the primary driver of alignment quality**. This result follows naturally from preference reconstruction (Section 2), but is unavailable under existing approaches, which lack an explicit notion of basis expressiveness. Beyond the core finding, the analysis reveals that our framework also serves as a diagnostic tool for alignment research. Prediction errors scale with cultural divergence of questions (Figure 3C), offering an empirical handle on learnability limits (Def. 2.2), while locale-matched generation gains its edge on culturally divergent dimensions (Figure 3D), suggesting that Stage 1 localization helps in tail cases. This empirical decomposition of alignment failure into basis expressiveness, model backend’s default cultural skew, and the cultural divergence of ground truths pertaining to the topic domain demonstrates that the novelty of our method goes beyond theoretical and practical grounds. Notably, the basis expressiveness that drives these results is characterized by entropy—P2P’s operational Stage 1 objective—further grounding this design choice empirically.
> > >
> > > Secondly, to build on our initial response on the technical contributions, we wish to clarify that while we acknowledged that persona prompting and regularized regression are individually established, **Stage 1 of P2P goes substantially beyond standard persona prompting**. As discussed in Section 3.2, the engineering goal of Stage 1 is to construct a diverse functional basis of agent preferences, but the prompt space is far too vast to explore in an unstructured manner (lines 181-182). P2P addresses this by treating attributes (factors that correlate with preference expression) as control handles for steering model behaviour, not as demographic descriptors for generating plausible personas. On this foundation, Stage 1 implements a structured generation system purpose-built to construct a diverse functional basis:
> > >
> > > - hierarchical and extensible attribute bank organized by template, mode, and attribute (Appendix B.1. and Figure 4);
> > > - LLM-driven attribute discovery from survey questions (Appendix B.2.1);
> > > - entropy-based adaptive sampling with bandit-style mode selection, question patching, and mixed-mode cross-pollination (Appendix B.3, Algorithms 1 & 2).
> > >
> > > The core generation logic is organized around **attributes** not personas, while the instantiation through endowment/persona is to aid generation coherence and interpretability to human readers (lines 188-191, footnote 8). **This integrated design is not an adaptation of existing persona generation or demographic conditioning pipelines**, and we are not aware of prior work implementing a similar logic.
> > >
> > > Moreover, we respectfully note that the reviewer’s concern rests on the premise that because persona prompting and L1-regularized regression are individually established, their combination offers limited innovation. Here we offer a counterpoint: a natural implementation of this premise—generating a large persona pool with diversity-instructed prompts, ensuring diversity further via K-means clustering on embeddings, then applying regression—does not improve performance but increases cost (Table 6). As discussed in Appendix D.1.1, clustering in embedding space tends to group personas by textual and stylistic similarity, which does not necessarily track preference-relevant variation. P2P succeeds because it optimizes for a different notion of diversity, grounded in revealed preferences rather than persona features.
> > >
> > > The reviewer states that aggregating personas via lasso or elastic net is “well known” in the literature. If the reviewer has specific references in mind, we would welcome them—this would help us better contextualize our contribution. We note that our simulation studies (Appendix C)—covering ground-truth recovery, reconstruction absent ground-truth basis, and the expressiveness-recoverability relationship—constitute, to our knowledge, the first systematic investigation of L1 regression as a preference reconstruction operator.
> > >
> > > We hope these observations, together with our initial rebuttal, clarify the nature and depth of our contribution.

---

### Decision · Program_Chairs · 2026-04-30

**Decision:**

Reject

**Comment:**

This paper studies the problem of aligning large language models with population-level human preferences and proposes modeling  population preference as a mixture of basis representative preferences which are learned from the data. The problem formulation of latent representations for capturing preferences as well as mixture of basis preferences have already been applied to pluralistic alignment problem and therefore is not novel. Additionally, as highlighted by the reviewers, the experiments do not compare with the relevant pluralistic preference learning methods. In summary, the submission is not sufficiently ready for ICML publication.